# NaRnEA: An Information Theoretic Framework for Gene Set Analysis

**DOI:** 10.3390/e25030542

**Published:** 2023-03-21

**Authors:** Aaron T. Griffin, Lukas J. Vlahos, Codruta Chiuzan, Andrea Califano

**Affiliations:** 1Medical Scientist Training Program, Columbia University Irving Medical Center, New York, NY 10032, USA; 2Department of Systems Biology, Columbia University Irving Medical Center, New York, NY 10032, USA; 3Department of Biostatistics, Columbia University Irving Medical Center, New York, NY 10032, USA; 4Department of Biochemistry and Molecular Biophysics, Columbia University, New York, NY 10032, USA; 5Department of Medicine, Vagelos College of Physicians and Surgeons, Columbia University, New York, NY 10032, USA; 6JP Sulzberger Columbia Genome Center, Columbia University Irving Medical Center, New York, NY 10032, USA; 7Department of Biomedical Informatics, Columbia University, New York, NY 10032, USA; 8Herbert Irving Comprehensive Cancer Center, Columbia University Irving Medical Center, New York, NY 10032, USA

**Keywords:** gene set analysis, principle of maximum entropy, nonparametric statistics, protein activity, regulatory networks

## Abstract

Gene sets are being increasingly leveraged to make high-level biological inferences from transcriptomic data; however, existing gene set analysis methods rely on overly conservative, heuristic approaches for quantifying the statistical significance of gene set enrichment. We created Nonparametric analytical-Rank-based Enrichment Analysis (NaRnEA) to facilitate accurate and robust gene set analysis with an optimal null model derived using the information theoretic Principle of Maximum Entropy. By measuring the differential activity of ~2500 transcriptional regulatory proteins based on the differential expression of each protein’s transcriptional targets between primary tumors and normal tissue samples in three cohorts from The Cancer Genome Atlas (TCGA), we demonstrate that NaRnEA critically improves in two widely used gene set analysis methods: Gene Set Enrichment Analysis (GSEA) and analytical-Rank-based Enrichment Analysis (aREA). We show that the NaRnEA-inferred differential protein activity is significantly correlated with differential protein abundance inferred from independent, phenotype-matched mass spectrometry data in the Clinical Proteomic Tumor Analysis Consortium (CPTAC), confirming the statistical and biological accuracy of our approach. Additionally, our analysis crucially demonstrates that the sample-shuffling empirical null models leveraged by GSEA and aREA for gene set analysis are overly conservative, a shortcoming that is avoided by the newly developed Maximum Entropy analytical null model employed by NaRnEA.

## 1. Introduction

Next-generation sequencing technologies and highly accurate annotations for prokaryotic and eukaryotic genomes have transformed biology into a data-rich scientific discipline [1]. Consequently, the field of computational biology has prioritized the development of algorithms that enable researchers to accurately leverage large-scale, gene-level biochemical measurements to make mechanistic inferences involving biological and cellular processes [2,3], as well as to measure the activity of molecular pathways and proteins [4,5]. It is not surprising that gene set analysis methods, which were developed to integrate statistical information from groups of genes belonging to a common ontology (e.g., biological process, metabolic pathway, regulatory network), have rapidly emerged as some of the most widely utilized tools in biomedical research. Most frequently, the differential expression of genes between two cellular states or phenotypes is used as the ranking criterion, though various other procedures may be employed. See (Maleki et al., 2020 Frontiers in Genetics) [6] and (Das et al., 2020 Entropy) [7] for recent reviews discussing the wide variety of published gene set analysis methods as well as the statistical assumptions implicit to each one.

While existing gene set analysis methods employ distinct mathematical approaches for calculating the test statistic associated with a gene set’s enrichment, the field of gene set analysis is uniquely dominated by the question of how to accurately evaluate the statistical significance of gene set enrichment. The origins of this debate are alluded to by Mootha et al. [8] in their analysis of DNA microarrays profiling the expression of genes in skeletal muscle biopsy samples from patients with normal glucose tolerance (NGT) or type 2 diabetes mellitus (DM2). Mootha et al. computed the differential expression of each gene in the test phenotype samples (i.e., the DM2 patients) with respect to the reference phenotype samples (i.e., the NGT patients) using the Signal-to-Noise Ratio (SNR). When this analysis failed to identify any single gene with statistically significant differential expression, Mootha et al. created a procedure they referred to as Gene Set Enrichment Analysis (GSEA); this method was designed to test the null hypothesis that the rank ordering of genes from a gene set in the differential gene expression signature (i.e., the vector of SNR values computed between the test phenotype and the reference phenotype) is random with regard to the diagnostic categorization of the samples.

Mootha et al. used a two-sample Kolmogorov–Smirnov test to compare the SNR values for genes associated with oxidative phosphorylation (OXPHOS) with the SNR values for all other genes. Rather than calculating the statistical significance of this enrichment score using the existing analytical null model for the two-sample Kolmogorov–Smirnov test, Mootha et al. chose to approximate the null model for their gene set enrichment score using an empirical phenotype-based permutation test procedure. In this procedure, Mootha et al. shuffled the phenotype label of each DM2 sample and NGT sample to produce two new groups: a null test phenotype and a null reference phenotype. Each of these null phenotypes consisted of samples from both the original test phenotype (i.e., patients with DM2) and the original reference phenotype (i.e., patients without DM2). They recomputed a null SNR for each gene based on its expression in the null test samples and the null reference samples, producing a null differential gene expression signature; the same two-sample Kolmogorov–Smirnov test statistic was then calculated as the null enrichment score for the OXPHOS gene set in this null differential gene expression signature. Mootha et al. repeated this procedure of permuting phenotype labels, calculating a null differential gene expression signature and computing a null enrichment score 1000 times, allowing them to estimate the statistical significance (i.e., two-sided *p*-value) of the OXPHOS gene set enrichment.

The logic of this empirical phenotype-based permutation null model for GSEA was more clearly described by Subramanian et al. [9] in their follow-up manuscript, in which GSEA was modified to use a weighted two-sample Kolmogorov–Smirnov test statistic, as follows:
*“We estimate the statistical significance (nominal P value) of the [GSEA enrichment score] by using an empirical phenotype-based permutation test procedure that preserves the complex correlation structure of the gene expression data. Specifically, we permute the phenotype labels and recompute the [GSEA enrichment score] of the gene set for the permuted data, which generates a null distribution for the [GSEA enrichment score]. The empirical, nominal P value of the observed [GSEA enrichment score] is then calculated relative to this null distribution. Importantly, the permutation of class labels preserves gene-gene correlations and, thus, provides a more biologically reasonable assessment of significance than would be obtained by permuting genes”.*

This justification for the empirical phenotype-based permutation null model of GSEA was defended by Tamayo et al. [10] when they compared GSEA with an alternative gene set analysis procedure, referred to by Tamayo et al. as Simple Enrichment Analysis (SEA), which relied on a null model for gene set analysis that ignores correlations between genes rather than the empirical phenotype-based permutation null model of GSEA. Tamayo et al. made the following claim about their comparison of GSEA and SEA:
*“We show, in agreement with earlier observations, that the gene independence assumption is not realistic because gene correlations are non-trivial and produce a substantial amount of variance inflation in the global statistic that in turn produces a large number of false positive results”.*

We can more clearly state the central claim underlying this procedure with symbolic logic as follows:

A → B

A: Genes in a gene set are correlated when the gene setis not enriched in the gene expression signature.B: A gene set analysis method that assumes independence betweengenes will not control the Type I error rate of gene set analysis.

Tamayo et al. sought to prove their primary claim (i.e., statement A) by showing that SEA produced a large number of false positives in their benchmark analyses (i.e., statement B). Unfortunately, such a verification would only be valid if the converse of the statement (i.e., B → A) were true in general, and the converse of a statement and the statement itself are not logically equivalent in general. However, the statement (A → B) does prove useful through its contrapositive, which may be stated as follows:

¬B → ¬A

¬B: A gene set analysis method that assumes independence betweengenes adequately controls the Type I error rate of gene set analysis.¬A: Genes in a gene set are not correlated when the gene setis not enriched in the gene expression signature.

Since the contrapositive of a statement and the statement itself are always logically equivalent, we find that the primary claim underlying the validity of the empirical phenotype-based permutation null model for GSEA can be falsified if we are able to create a gene set analysis method that assumes that genes in a gene set are independent when the gene set is not enriched in a gene expression signature and subsequently show that this method adequately controls the Type I error rate of gene set analysis. Additionally, we note that the approach undertaken by Tamayo et al. as an attempt to defend GSEA and falsify SEA is fundamentally flawed because the method that Tamayo et al. used to evaluate the specificity of GSEA and SEA is the same empirical phenotype-based permutation procedure that GSEA relies on to estimate the null model for gene set enrichment. Tamayo et al. describe their benchmarking procedure as follows:
*“For each dataset in the benchmark, we randomized the phenotype labels 1000 times and ran both algorithms … The p values are computed using the areas under the empirical null histograms from GSEA and areas under the normal distribution for SEA”.*

We see this is the same description provided by Subramanian et al. [9] for the empirical phenotype-based permutation procedure used to construct the null model for GSEA; thus, Tamayo et al. have attempted to defend the sample-shuffling null model of GSEA by way of tautology, rendering their argument invalid.

This discussion also serves to highlight the challenges intrinsic to accurately benchmarking gene set analysis methods using experimental data; in particular, gene sets that are frequently analyzed are often derived from the literature for biological processes or other gene ontologies that are not amenable to systematic, experimental validation. However, recent work in the field of cancer systems biology has shown that gene set analysis methods can be used to measure the differential activity of transcriptional regulatory proteins [4,11]; in the same way that the Michaelis–Menten equation measures enzymatic activity based on the conversion of biochemical substrates to metabolic products, we may interpret the differential expression of a transcriptional regulator’s targets as an estimate of that regulator’s differential activity. More formally, we define the differential activity of a transcriptional regulatory protein as the contribution of the regulator to the implementation of a specific differential gene expression signature. Consistent with this definition, and akin to using a highly multiplexed gene reporter assay, we previously introduced the Virtual Inference of Protein Activity by Enriched Regulon Analysis (VIPER) algorithm to measure differential protein activity based on the enrichment of each regulator’s transcriptional targets (i.e., regulon gene set) in a differential gene expression signature [4]. The tissue-specific regulon gene sets required for these analyses can be effectively reverse-engineered using a variety of methods [12]; in the context of this study, we use the ARACNe3 algorithm, which is the newest implementation of the Algorithm for the Reconstruction of Accurate Cellular Networks (ARACNe) [13,14]. Previous versions of this algorithm have been experimentally validated, and the regulon gene sets created by ARACNe have been used extensively to measure the differential activity of transcriptional regulatory proteins in combination with VIPER, effectively identifying Master Regulator proteins representing mechanistic determinants of tumor transcriptional states [11,15,16].

In this manuscript, we derive and benchmark Nonparametric analytical-Rank-based Enrichment Analysis (NaRnEA), a novel gene set analysis method that leverages a fully analytical null model for gene set enrichment created using the information theoretic Principle of Maximum Entropy [17]. By virtue of its derivation, the null model for NaRnEA assumes that genes in a gene set are independent when the gene set is not enriched in a gene expression signature. We show that NaRnEA adequately controls the Type I error rate of gene set analysis using gene expression data from the lung adenocarcinoma (LUAD), colon adenocarcinoma (COAD), and head and neck squamous cell carcinoma (HNSC) cohorts in The Cancer Genome Atlas (TCGA). Our finding that NaRnEA adequately controls the Type I error rate of gene set analysis effectively falsifies the primary claim underlying the empirical phenotype-based permutation null model of GSEA.

Furthermore, we demonstrate that NaRnEA is highly sensitive, identifying far more statistically significantly enriched regulon gene sets in these TCGA cohorts than either GSEA or analytical-Rank-based Enrichment Analysis (aREA), the gene set analysis method originally developed as the computational engine of VIPER that also employs an empirical phenotype-based permutation null model. Using independent proteomic data from the Clinical Proteomic Tumor Analysis Consortium (CPTAC) for LUAD, COAD, and HNSC cancer types, we demonstrate that the differential activity of transcriptional regulatory proteins measured by NaRnEA in TCGA is significantly correlated with the differential abundance of the same transcriptional regulatory proteins in CPTAC. Given that the abundance and the activity of transcriptional regulatory proteins differ due to a number of biochemical processes (e.g., post-translational modification, subcellular localization, cofactor binding, chromatin accessibility), this agreement provides substantial biological support for NaRnEA-inferred differential protein activity. Crucially, this comparative analysis is not possible in a large-scale, systematic fashion for literature-curated gene sets since the corresponding gene ontologies are infrequently amenable to independent, experimental validation.

These findings demonstrate that NaRnEA is statistically robust, having greater sensitivity than either GSEA or aREA without loss of specificity, and produces biologically meaningful inferences. We also interrogated the statistical properties of the empirical phenotype-based permutation procedure leveraged by GSEA and aREA and determined that the resulting null gene expression signatures exhibit substantial correlation with the true gene expression signature, thus providing a rigorous explanation for the reduced sensitivity of any gene set analysis method that relies on this procedure to approximate the null model for gene set enrichment. Finally, we identify systematic flaws in both GSEA and aREA when these methods are applied using alternative null models for gene set analysis while further highlighting the excellent performance of NaRnEA. NaRnEA and ARACNe3, along with all the code necessary to reproduce these analyses, are freely available for research use on GitHub (https://github.com/califano-lab/NaRnEA (accessed on 3 March 2023)).

## 2. Materials and Methods

### 2.1. Nonparametric Analytical-Rank-Based Enrichment Analysis (NaRnEA)

We begin our derivation of NaRnEA by first considering two phenotypes, which we will refer to as the test phenotype (A) and the reference phenotype (B). Let us assume that there is some gene (g) and that we may represent the expression of this gene using the discrete random variable (Xg); biochemically speaking, this is a compositional discrete random variable that represents the relative molar concentration of transcripts originating from the *g*th genomic locus. If we would like to be more specific, we can say that we are representing the expression of the gene in the test phenotype (A) with the discrete random variable (XgA) and we are representing the expression of the gene in the reference phenotype (B) with the discrete random variable (XgB). To perform gene set analysis, we must first determine whether the *g*th gene is more highly expressed in the test phenotype (A) or the reference phenotype (B); this calculation forms the basis of our differential gene expression signature. To remain as general as possible, let us assume that we may represent the differential gene expression signature value for the *g*th gene between the test phenotype (A) and the reference phenotype (B) with the value (zgAB). We will assume in the following derivation that (zgAB) has the following properties:zgAB ϵ ℝWe assume that the differential gene expression signature value for the *g*th gene between the test phenotype (A) and the reference phenotype (B) is a real number that may be positive or negative.zgAB>0 iff Pr(XgA>XgB)>12We assume that the differential gene expression signature value for the *g*th gene between the test phenotype (A) and the reference phenotype (B) is greater than zero if and only if the expression of the *g*th gene in the test phenotype (A) is greater than the expression of the *g*th gene in the reference phenotype (B). More formally, this may be expressed by stating that the discrete random variable that represents the expression of the *g*th gene in the test phenotype (A) stochastically dominates the discrete random variable that represents the expression of the *g*th gene in the reference phenotype (B).zgAB<0 iff Pr(XgA>XgB)<12We assume that the differential gene expression signature value for the *g*th gene between the test phenotype (A) and the reference phenotype (B) is less than zero if and only if the expression of the *g*th gene in the test phenotype (A) is less than the expression of the *g*th gene in the reference phenotype (B). More formally, this may be expressed by stating that the discrete random variable that represents the expression of the *g*th gene in the test phenotype (A) is stochastically dominated by the discrete random variable that represents the expression of the *g*th gene in the reference phenotype (B).zgAB=−zgBAWe assume that the differential gene expression signature value for the *g*th gene between the test phenotype (A) and the reference phenotype (B) will be equal in magnitude and opposite in sign if the ordering of the phenotypes were reversed; this follows naturally from the aforementioned definitions of positive differential gene expression (i.e., upregulation) and negative differential gene expression (i.e., downregulation).|zgAB|>|zkAB| iff |Pr(XgA>XgB)−12|>|Pr(XkA>XkB)−12| for g≠kWe assume that the magnitude of the differential gene expression signature value for the *g*th gene between the test phenotype (A) and the reference phenotype (B) should be greater than the magnitude of the differential gene expression signature value for the *k*th gene between the test phenotype (A) and the reference phenotype (B) if and only if the extent of differential expression for the *g*th gene is greater than the extent of differential expression for the *k*th gene between the two phenotypes; we formalize this notion here using the language of stochastic dominance as mentioned previously.

Having considered the differential gene expression signature, we may now turn our attention to the gene set itself. So long as the gene set members are determined a priori, the biological rationale underlying the gene set’s construction is irrelevant for the statistical analysis of the gene set’s enrichment in the differential gene expression signature; however, it plays a crucial role in the biological interpretation of the enrichment, if it is indeed present. To formalize the notion of gene set analysis from first principles, we may select a gene set for which the expression of each member exhibits a statistical dependency on a common biochemical species; from this we may construct a conceptual rationale for gene set analysis and derive an appropriate mathematical framework that will facilitate accurate statistical inference.

We assume, for the sake of this derivation, that a transcriptional regulatory protein (r) is responsible for regulating the expression of the gene (g). To formalize this relationship, let us assume that we may represent the activity of the regulator (r) using the discrete random variable ($Y_{r}$); biochemically speaking, this is a compositional discrete random variable that represents the relative molar concentration of the transcriptional regulatory holoenzyme. Then, due to the statistical dependence induced by this biochemical relationship, the random variables ($Y_{r}$) and ($X_{g}$) form the following Markov Chain (Equation (1)):(1)$$Y_{r}\to X_{g}$$

If the *r*th regulatory protein regulates many genes, it would be helpful to distinguish between these different targets based on the strength of each regulatory relationship. We can quantify the degree to which the expression of the *g*th gene depends on the activity of the *r*th regulator using a parameter to which we refer as the Association Weight ($AW_{rg}$). We require the Association Weight to have the following properties:$AW_{rg}\geq 0$The Association Weight between the *r*th regulator and the *g*th gene is strictly non-negative.$AW_{rg}>0\;iff\;I\left[Y_{r};X_{g}\right]>0$The Association Weight between the *r*th regulator and the *g*th gene is greater than zero if and only if the expression of the *g*th gene exhibits a statistical dependency on the activity of the *r*th regulator; this may be formalized by stating that the mutual information between the discrete random variable representing the expression of the *g*th gene and the discrete random variable representing the activity of the *r*th regulator is nonzero.$AW_{rg}>AW_{rk}\;iff\;I\left[Y_{r};X_{g}\right]>I\left[Y_{r};X_{k}\right]\;for\;g\neq k$The Association Weight between the *r*th regulator and the *g*th gene is greater than the Association Weight between the *r*th regulator and the *k*th gene if and only if the expression of the *g*th gene exhibits a greater statistical dependency on the activity of the *r*th regulator than the expression of the *k*th gene as measured using the mutual information between the corresponding discrete random variables.

In addition to characterizing the regulatory relationships between the *r*th regulatory protein and its targets based on their strength, we can describe them based on their directionality. In some cases, as the activity of a regulator increases, it may cause the expression of some targets to increase; in other cases, an increase in the activity of the regulator may cause the expression of some targets to decrease. We quantify the degree to which the expression of the *g*th gene increases or decreases monotonically based on an increase in the activity of the *r*th regulator using a parameter to which we refer as the Association Mode ($AM_{rg}$). We require the Association Mode to have the following properties:$AM_{rg}\;\in \left[-1,\;1\right]$The Association Mode between the *r*th regulator and the *g*th gene is a real number less than or equal to one and greater than or equal to negative one.$AM_{rg}>0\;iff\;SCC\left[Y_{r},X_{g}\right]>0$The Association Mode between the *r*th regulator and the *g*th gene is greater than zero if and only if there is a positive monotonic relationship between the activity of the *r*th regulator and the expression of the *g*th gene; this may be formalized by stating that the Spearman correlation coefficient between the discrete random variable representing the expression of the *g*th gene and the discrete random variable representing the activity of the *r*th regulator is positive.$AM_{rg}<0\;iff\;SCC\left[Y_{r},X_{g}\right]<0$The Association Mode between the *r*th regulator and the *g*th gene is less than zero if and only if there is a negative monotonic relationship between the activity of the *r*th regulator and the expression of the *g*th gene; this may be formalized by stating that the Spearman correlation coefficient between the discrete random variable representing the expression of the *g*th gene and the discrete random variable representing the activity of the *r*th regulator is negative.$\left|AM_{rg}\right|>\left|AM_{rk}\right|\;iff\;\left|SCC\left[Y_{r},X_{g}\right]\right|>\left|SCC\left[Y_{r},X_{k}\right]\right|\;for\;g\neq k$The magnitude of the Association Mode between the *r*th regulator and the *g*th gene is greater than the magnitude of the Association Mode between the *r*th regulator and the *k*th gene if and only if the Spearman correlation coefficient between the expression of the *g*th gene and the activity of the *r*th regulator is greater in magnitude than the Spearman correlation coefficient between the expression of the *kth* gene and the activity of the *r*th regulator quantified from the corresponding discrete random variables.

Given that we have now parameterized the relationship between the *r*th regulator and its transcriptional targets using the Association Weight and Association Mode, we can formalize the notion of regulon gene set enrichment in a differential gene expression signature resulting from a change in the activity of the regulator. More specifically, let ($Y_{rA}$) be the discrete random variable that represents the activity of the *r*th regulator in the test phenotype (A), and let ($Y_{rB}$) be the discrete random variable that represents the activity of the *r*th regulator in the reference phenotype (B). These discrete random variables may be related in one of three ways:$\mathrm{Pr}\left(Y_{rA}>Y_{rB}\right)>\frac{1}{2}$In the first scenario, the activity of the *r*th regulator in the test phenotype (A) is greater than the activity of the *r*th regulator in the reference phenotype (B). More formally, this may be expressed by stating that the discrete random variable that represents the activity of the *r*th regulator in the test phenotype (A) stochastically dominates the discrete random variable that represents the activity of the *r*th regulator in the reference phenotype (B).$\mathrm{Pr}\left(Y_{rA}>Y_{rB}\right)<\frac{1}{2}$In the second scenario, the activity of the *r*th regulator in the test phenotype (A) is less than the activity of the *r*th regulator in the reference phenotype (B). More formally, this may be expressed by stating that the discrete random variable that represents the activity of the *r*th regulator in the test phenotype (A) is stochastically dominated by the discrete random variable that represents the activity of the *r*th regulator in the reference phenotype (B).$\mathrm{Pr}\left(Y_{rA}>Y_{rB}\right)=\frac{1}{2}$In the third scenario, the activity of the *r*th regulator in the test phenotype (A) is equal to the activity of the *r*th regulator in the reference phenotype (B). More formally, this may be expressed by stating that the discrete random variable that represents the activity of the *r*th regulator in the test phenotype (A) neither stochastically dominates nor is stochastically dominated by the discrete random variable that represents the activity of the *r*th regulator in the reference phenotype (B).

We refer to scenario 1 as our positive alternative hypothesis ($H_{a}^{+}$), scenario 2 as our negative alternative hypothesis ($H_{a}^{-}$), and scenario 3 as our null hypothesis ($H_{o}$). As we have stated previously, the primary statistical concern in the field of gene set analysis has been correctly defining the joint sampling distribution of gene set members in the differential gene expression signature when the null hypothesis is true. In order to construct the null model for NaRnEA, we begin by considering the joint sampling distribution of the gene set members in the differential gene expression signature when each form of the alternative hypothesis is true.

Under the positive alternative hypothesis ($H_{a}^{+}$), the activity of the *r*th regulator in the test phenotype (A) is greater than the activity of the *r*th regulator in the reference phenotype (B). It follows from our earlier discussion about the gene set parameters that the genes with a nonzero Association Weight ($AW_{rg}$) will exhibit differential expression between the test phenotype (A) and the reference phenotype (B), such that $z_{gAB}$ will be nonzero. Furthermore, a gene with a positive Association Mode ($AM_{rg}>0$) will have a positive differential gene expression signature value ($z_{gAB}>0$), and a gene with a negative Association Mode ($AM_{rg}<0$) will have a negative differential gene expression signature value ($z_{gAB}<0$).

Under the negative alternative hypothesis ($H_{a}^{-}$), the activity of the *r*th regulator in the test phenotype (A) is less than the activity of the *r*th regulator in the reference phenotype (B). It follows from our earlier discussion about the gene set parameters that the genes with a nonzero Association Weight ($AW_{rg}$) will exhibit differential expression between the test phenotype (A) and the reference phenotype (B), such that $z_{gAB}$ will be nonzero. Furthermore, a gene with a positive Association Mode ($AM_{rg}>0$) will have a negative differential gene expression signature value ($z_{gAB}<0$), and a gene with a negative Association Mode ($AM_{rg}<0$) will have a positive differential gene expression signature value ($z_{gAB}>0$).

We find that either form of the alternative hypothesis implies that the joint sampling distribution of the gene set members has greater probability density at the extremes of the differential gene expression signature than near the center of the differential gene expression signature; whether that increase occurs at the positive extreme or negative extreme for a given gene depends on the version of the alternative hypothesis for its regulator and the Association Mode for that gene. Additionally, the degree to which the probability mass increases at the extremes for a particular gene depends on the Association Weight for the gene as well as the magnitude of differential activity for its regulator.

Our analysis of the joint sampling distribution for the gene set members can be greatly simplified if we apply a nonparametric transformation to the differential gene expression signature, as follows (Equation (2)):(2)$${\left\{z_{g}\right\}}_{AB}⟼{\left\{r_{g}s_{g}\right\}}_{AB}\;r_{gAB}=rank\left(\left|z_{gAB}\right|\right)\;s_{gAB}=sign\left(z_{gAB}\right)$$

As a result of the nonparametric transformation in Equation (2), we can instead consider the discrete joint sampling distribution for the gene set members where the domain of each marginal is $\left\{r_{1AB}s_{1AB},\;\ldots ,\;r_{GAB}s_{GAB}\right\}$. If we let
(3)$$N_{r}=\overset{G}{\underset{g=1}{\sum }}I\left\{AW_{rg}>0\right\}$$
be the number of genes in the regulon gene set of the *r*th regulator, where I{·} is the indicator function, then the discrete joint sampling distribution has the dimensionality ($N_{r}\times G$).

Subsequently, we recognize that from an information theoretic perspective, both versions of the alternative hypothesis reduce the Shannon entropy [18] of the discrete joint sampling distribution for the gene set members due to an increase in the probability mass at extremes of the nonparametric differential gene expression signature. Furthermore, if we consider the magnitude of the difference in the protein activity between the test phenotype (A) and the reference phenotype (B), we conclude that a greater difference in the activity of the *r*th regulator corresponds with a greater increase in probability mass at the extremes of the nonparametric differential gene expression signature and therefore a more substantial reduction in the joint Shannon entropy of the discrete joint probability distribution. It follows from this line of reasoning that when the magnitude of the difference in the activity of the *r*th regulator between the test phenotype (A) and the reference phenotype (B) tends to zero, the joint Shannon entropy of the discrete joint probability distribution for the members of the corresponding regulon gene set in the nonparametric differential gene expression signature will tend to its largest possible value. Therefore, we invoke the Principle of Maximum Entropy to motivate our selection of the discrete joint sampling distribution with the greatest Shannon entropy in the nonparametric differential gene expression signature as the null distribution for the gene set members.

To derive the Maximum Entropy null model for the gene set members in the nonparametric differential gene expression signature, we first consider the gene expression marginals, each of which constitutes a discrete probability distribution over (G) elements. Under the null hypothesis, the only information available to us is that each gene set member is present somewhere in the nonparametric differential gene expression signature; beyond this, we have no additional information about the expected value or higher moments (e.g., variance, skewness) of the marginal distribution under the null hypothesis. Thus, it follows from the log-sum inequality [19] that the entropy of each gene expression marginal is maximized when we assign equal probability mass to all possible elements of the nonparametric differential gene expression signature for each gene under the null hypothesis. This is equivalent to assuming that each member of the gene set is uniformly distributed in the nonparametrically transformed differential gene expression signature under the null hypothesis of gene set analysis.

Furthermore, it follows from the log-sum inequality that for an ensemble of discrete random variables, the entropy of the joint probability distribution is always less than or equal to the sum of the univariate entropies with equality if and only if the discrete random variables that compose the ensemble are statistically independent. Thus, to maximize the joint entropy of the null model, we assume that the gene set members are independent and uniformly distributed in the nonparametric differential gene expression signature under the null hypothesis of gene set analysis. We note that this Maximum Entropy null model for NaRnEA contradicts the primary claim underlying the validity of the empirical phenotype-based permutation null model for GSEA (i.e., that genes in a gene set are correlated when the gene set is not enriched in the differential gene expression signature). Thus, to falsify this claim, it will be sufficient to demonstrate that NaRnEA is capable of adequately controlling the Type I error rate of gene set analysis.

To quantify the extent to which the regulon gene set for the *r*th regulator is enriched in the nonparametric differential gene expression signature computed between the test phenotype (A) and the reference phenotype (B), NaRnEA leverages two complementary test statistics—the Directed Enrichment Score ($DES_{rAB}$) and the Undirected Enrichment Score ($UES_{rAB}$)—which are defined as follows (Equation (4)):(4)$$DES_{rAB}=\underset{g}{\sum }DES_{rgAB}\;DES_{rgAB}=\left(AW_{rg}\right)\left(AM_{rg}\right)\left(r_{gAB}s_{gAB}\right)\;UES_{rAB}=\underset{g}{\sum }UES_{rgAB}\;UES_{rgAB}=\left(AW_{rg}\right)\left(1-\left|AM_{rg}\right|\right)\left(r_{gAB}\right)$$

Both the Directed Enrichment Score and Undirected Enrichment Score weight the contribution of each gene toward the enrichment of the gene set based on the Association Weight parameter since the differential expression of a gene whose expression depends more strongly on the activity of the regulator is a better indicator of the change in the regulator’s activity. However, these complementary test statistics differ in how each incorporates the Association Mode. The Directed Enrichment Score considers both the magnitude and sign of the differential expression for each gene set member, whereas the Undirected Enrichment Score considers only the magnitude of differential expression. It follows that gene set members that are monotonically regulated by the *r*th regulator should contribute more to the Directed Enrichment Score since the sign of their differential gene expression signature values will be important for determining whether the null hypothesis of gene set analysis ought to be rejected in favor of the positive alternative hypothesis or the negative alternative hypothesis. However, gene set members that are non-monotonically regulated by the *r*th regulator should not contribute substantially to the Directed Enrichment Score since the sign of their differential expression does not clearly support one version of the alternative hypothesis over another. To that end, the formulation of the Undirected Enrichment Score provides a mechanism by which the non-monotonically regulated members of the regulon gene set may contribute to the enrichment.

Since the Directed Enrichment Score and Undirected Enrichment score for the regulon gene set of the *r*th regulator are computed from the same gene set members, they form a bivariate vector $\left\{DES_{rAB},UES_{rAB}\right\}$. We recognize that each of these test statistics is equal to the sum of independent random variables under the null hypothesis of gene set analysis, allowing us to invoke the multivariate version of the Lindeberg Central Limit Theorem [20] to derive the asymptotic null distribution of this bivariate vector. More formally, we define the Normalized Directed Enrichment Score ($NDES_{rAB}$) for the regulon gene set of the *r*th regulator in the nonparametric differential gene expression signature computed between the test phenotype (A) and the reference phenotype (B) as follows (Equations (5)–(7)):(5)$$NDES_{rAB}=\frac{DES_{rAB}-E\left[DES_{rAB}|H_{o}\right]}{\sqrt{V𝕒𝕣\left[DES_{rAB}|H_{o}\right]}}$$
(6)$$E\left[DES_{rAB}|H_{o}\right]=\overset{G}{\underset{g=1}{\sum }}E\left[DES_{rgAB}|H_{o}\right]\;\;E\left[DES_{rgAB}|H_{o}\right]=\left(AW_{rg}\right)\left(AM_{rg}\right)E\left[r_{gAB}s_{gAB}|H_{o}\right]\;E\left[r_{gAB}s_{gAB}|H_{o}\right]=\overset{G}{\underset{k=1}{\sum }}\left(r_{kAB}s_{kAB}\right)\times \mathrm{Pr}\left(r_{gAB}s_{gAB}=r_{kAB}s_{kAB}|H_{o}\right)=\frac{1}{G}\overset{G}{\underset{k=1}{\sum }}r_{kAB}s_{kAB}$$
(7)$$V𝕒𝕣\left[DES_{rAB}|H_{o}\right]=\overset{G}{\underset{g=1}{\sum }}V𝕒𝕣\left[DES_{rgAB}|H_{o}\right]\;V𝕒𝕣\left[DES_{rgAB}|H_{o}\right]=E\left[DES_{rgAB}^{2}|H_{o}\right]-E{\left[DES_{rgAB}|H_{o}\right]}^{2}\;E\left[DES_{rgAB}^{2}|H_{o}\right]={\left(AW_{rg}\right)}^{2}{\left(AM_{rg}\right)}^{2}E\left[r_{gAB}^{2}|H_{o}\right]\;E\left[r_{gAB}^{2}|H_{o}\right]=\overset{G}{\underset{k=1}{\sum }}\left(r_{kAB}^{2}\right)\times \mathrm{Pr}\left(r_{gAB}^{2}=r_{kAB}^{2}|H_{o}\right)=\frac{1}{G}\overset{G}{\underset{k=1}{\sum }}r_{kAB}^{2}=\left(\frac{1}{6}\right)\left(2G^{2}+3G+1\right)$$

Then, if the condition
(8)$$\underset{G\to \infty }{\lim}\frac{\sum_{g=1}^{G}E\left[\;{\left(DES_{rgAB}-E\left[DES_{rgAB}|H_{o}\right]\right)}^{2}\times I\left\{\left|DES_{rgAB}-E\left[DES_{rgAB}|H_{o}\right]\right|>\varepsilon \sqrt{V𝕒𝕣\left[DES_{rAB}|H_{o}\right]}\right\}\;|\;H_{o}\;\right]}{V𝕒𝕣\left[DES_{rAB}|H_{o}\right]}$$
is satisfied for all (ε>0) where I{·} is the indicator function, the Central Limit Theorem holds such that
(9)$$p\left(NDES_{rAB}|H_{o}\right)\overset{D}{\to }\;N\left(0,1\right)$$
and the Normalized Directed Enrichment Score for the regulon gene set of the *r*th regulator in the nonparametric differential gene expression signature computed between the test phenotype (A) and the reference phenotype (B) converges in distribution to a standard normal random variable under the null hypothesis of gene set analysis.

Similarly, we define the Normalized Undirected Enrichment Score ($NUES_{rAB}$) for the regulon gene set of the *r*th regulator in the nonparametric differential gene expression signature computed between the test phenotype (A) and the reference phenotype (B) as follows (Equations (10)–(12)):(10)$$NUES_{rAB}=\frac{UES_{rAB}-E\left[UES_{rAB}|H_{o}\right]}{\sqrt{V𝕒𝕣\left[UES_{rAB}|H_{o}\right]}}$$
(11)$$E\left[UES_{rAB}|H_{o}\right]=\overset{G}{\underset{g=1}{\sum }}E\left[UES_{rgAB}|H_{o}\right]\;E\left[UES_{rgAB}|H_{o}\right]=\left(AW_{rg}\right)\left(1-\left|AM_{rg}\right|\right)E\left[r_{gAB}|H_{o}\right]\;E\left[r_{gAB}|H_{o}\right]=\overset{G}{\underset{k=1}{\sum }}\left(r_{kAB}\right)\times \mathrm{Pr}\left(r_{gAB}=r_{kAB}|H_{o}\right)=\frac{1}{G}\overset{G}{\underset{k=1}{\sum }}r_{kAB}=\left(\frac{1}{2}\right)\left(G+1\right)$$
(12)$$V𝕒𝕣\left[UES_{rAB}|H_{o}\right]=\overset{G}{\underset{g=1}{\sum }}V𝕒𝕣\left[UES_{rgAB}|H_{o}\right]\;V𝕒𝕣\left[UES_{rgAB}|H_{o}\right]=E\left[UES_{rgAB}^{2}|H_{o}\right]-E{\left[UES_{rgAB}|H_{o}\right]}^{2}\;E\left[UES_{rgAB}^{2}|H_{o}\right]={\left(AW_{rg}\right)}^{2}{\left(1-\left|AM_{rg}\right|\right)}^{2}E\left[r_{gAB}^{2}|H_{o}\right]\;E\left[r_{gAB}^{2}|H_{o}\right]=\overset{G}{\underset{k=1}{\sum }}\left(r_{kAB}^{2}\right)\times \mathrm{Pr}\left(r_{gAB}^{2}=r_{kAB}^{2}|H_{o}\right)=\frac{1}{G}\overset{G}{\underset{k=1}{\sum }}r_{kAB}^{2}=\left(\frac{1}{6}\right)\left(2G^{2}+3G+1\right)$$

Then, if the condition
(13)$$\underset{G\to \infty }{\lim}\frac{\sum_{g=1}^{G}E\left[\;{\left(UES_{rgAB}-E\left[UES_{rgAB}|H_{o}\right]\right)}^{2}\times I\left\{\left|UES_{rgAB}-E\left[UES_{rgAB}|H_{o}\right]\right|>\varepsilon \sqrt{V𝕒𝕣\left[UES_{rAB}|H_{o}\right]}\right\}\;|\;H_{o}\;\right]}{V𝕒𝕣\left[UES_{rAB}|H_{o}\right]}$$
is satisfied for all (ε>0) where I{·} is the indicator function, the Central Limit Theorem holds and
$$p\left(NUES_{rAB}|H_{o}\right)\overset{D}{\to }\;N\left(0,1\right)$$
such that the Normalized Undirected Enrichment Score for the regulon gene set of the *r*th regulator in the nonparametric differential gene expression signature computed between the test phenotype (A) and the reference phenotype (B) converges in distribution to a standard normal random variable under the null hypothesis of gene set analysis.

We note that these sufficiency conditions, derived by Lindeberg, will be satisfied by any gene set with enough members (i.e., at least 30 targets) for which the Association Weight and Association Mode parameters are sufficiently well balanced; this ensures that the variance of the summand is not dominated by the variance of any element of the summand under the null hypothesis.

Thus, the vector that consists of the Normalized Directed Enrichment Score and the Normalized Undirected Enrichment Score is a bivariate normal random vector under the null hypothesis of gene set analysis; furthermore, the mean of each marginal is equal to zero, and the variance of equal marginal is equal to one. We can also compute the covariance of this bivariate normal random vector under the null hypothesis of gene set analysis as follows (Equation (14)):(14)$$C𝕠𝕧\left[NDES_{rAB},NUES_{rAB}|H_{o}\right]=\rho \left[NDES_{rAB},NUES_{rAB}|H_{o}\right]\;\rho \left[NDES_{rAB},NUES_{rAB}|H_{o}\right]=\rho \left[DES_{rAB},UES_{rAB}|H_{o}\right]\;\rho \left[DES_{rAB},UES_{rAB}|H_{o}\right]=\frac{C𝕠𝕧\left[DES_{rAB},UES_{rAB}|H_{o}\right]}{\sqrt{V𝕒𝕣\left[DES_{rAB}|H_{o}\right]\;V𝕒𝕣\left[UES_{rAB}|H_{o}\right]}}\;C𝕠𝕧\left[DES_{rAB},UES_{rAB}|H_{o}\right]=\overset{G}{\underset{g=1}{\sum }}C𝕠𝕧\left[DES_{rgAB},UES_{rgAB}|H_{o}\right]\;C𝕠𝕧\left[DES_{rgAB},UES_{rgAB}|H_{o}\right]=E\left[DES_{rgAB}UES_{rgAB}|H_{o}\right]-E\left[DES_{rgAB}|H_{o}\right]\;E\left[UES_{rgAB}|H_{o}\right]\;E\left[DES_{rgAB}UES_{rgAB}|H_{o}\right]={\left(AW_{rg}\right)}^{2}\left(AM_{rg}\right)\left(1-\left|AM_{rg}\right|\right)E\left[r_{gAB}^{2}s_{gAB}|H_{o}\right]\;E\left[r_{gAB}^{2}s_{gAB}|H_{o}\right]=\overset{G}{\underset{k=1}{\sum }}\left(r_{kAB}^{2}s_{kAB}\right)\times \mathrm{Pr}\left(r_{gAB}^{2}s_{gAB}=r_{kAB}^{2}s_{kAB}|H_{o}\right)=\frac{1}{G}\overset{G}{\underset{k=1}{\sum }}r_{kAB}^{2}s_{kAB}$$
where ρ[X,Y] is the Pearson product moment correlation between the random variables (X) and (Y).

To determine how we should interpret the bivariate vector consisting of the Normalized Directed Enrichment Score and the Normalized Undirected Enrichment Score as providing evidence for either positive or negative gene set enrichment in the nonparametric differential gene expression signature, we return to our previous discussion regarding the differential gene expression signature values for the members of the regulon gene set for the *r*th regulator under each version of the alternative hypothesis. Since each gene in the regulon gene set for the *r*th regulator with a positive Association Mode will have a positive differential gene expression signature value under the positive alternative hypothesis, and each gene in the regulon gene set for the *r*th regulator with a negative Association Mode will have a negative differential gene expression signature value under the positive alternative hypothesis, it follows that both the Normalized Directed Enrichment Score and the Normalized Undirected Enrichment Score will be positive under the positive alternative hypothesis. Based on this rationale, we can combine the Normalized Directed Enrichment Score with the Normalized Undirected Enrichment Score to produce a single test statistic that will be strongly positive under the positive alternative hypothesis, which we refer to as the positive Normalized Enrichment Score (Equation (15)):(15)$$NES_{rAB}^{+}=\frac{NDES_{rAB}+NUES_{rAB}}{\sqrt{2+2C𝕠𝕧\left[NDES_{rAB},NUES_{rAB}|H_{o}\right]}}$$

Since each gene in the regulon gene set for the *r*th regulator with a positive Association Mode will have a negative differential gene expression signature value under the negative alternative hypothesis, and each gene in the regulon gene set for the *r*th regulator with a negative Association Mode will have a positive differential gene expression signature value under the negative alternative hypothesis, it follows that the Normalized Directed Enrichment Score will be negative under the negative alternative hypothesis while the Normalized Undirected Enrichment Score will be positive under the negative alternative hypothesis. Based on this rationale, we can combine the Normalized Directed Enrichment Score with the Normalized Undirected Enrichment Score to produce a single test statistic that will be strongly negative under the negative alternative hypothesis, which we refer to as the negative Normalized Enrichment Score (Equation (16)):(16)$$NES_{rAB}^{-}=\frac{NDES_{rAB}-NUES_{rAB}}{\sqrt{2-2C𝕠𝕧\left[NDES_{rAB},NUES_{rAB}|H_{o}\right]}}$$

This biologically motivated change of variables is mathematically equivalent to an affine transformation of the original bivariate vector $\left\{NDES_{rAB},NUES_{rAB}\right\}\mapsto \left\{NES_{rAB}^{+},NES_{rAB}^{-}\right\}$. It is well established that an affine transformation of a multivariate normal random vector produces a new multivariate normal random vector whose mean vector and covariance matrix can be immediately calculated [21]. More formally, if (Y=BX+c) is an affine transformation of the multivariate normal random vector (X) with a mean vector equal to ($\mu_{X}$) and a covariance matrix equal to ($\Sigma_{X}$), then the random vector (Y) is a multivariate normal random vector with a mean vector equal to ($\mu_{Y}=B\mu_{x}+c$) and a covariance matrix equal to ($\Sigma_{Y}=B\Sigma_{X}B^{T}$). Letting ($\varphi =C𝕠𝕧\left[NDES_{rAB},NUES_{rAB}|H_{o}\right]$), we can calculate the mean vector and covariance matrix of our new bivariate vector under the null hypothesis of gene set analysis as follows (Equations (17)–(19)):(17)$$\left(\begin{array}{c} E\left[NES_{rAB}^{+}|H_{o}\right]\\ E\left[NES_{rAB}^{-}|H_{o}\right] \end{array}\right)=\left(\begin{array}{cc} \frac{1}{\sqrt{2+2\varphi }} & \frac{1}{\sqrt{2+2\varphi }}\\ \frac{1}{\sqrt{2-2\varphi }} & \frac{-1}{\sqrt{2-2\varphi }} \end{array}\right)\left(\begin{array}{c} E\left[NDES_{rAB}|H_{o}\right]\\ E\left[NUES_{rAB}|H_{o}\right] \end{array}\right)+\left(\begin{array}{c} 0\\ 0 \end{array}\right)\;\left(\begin{array}{c} E\left[NES_{rAB}^{+}|H_{o}\right]\\ E\left[NES_{rAB}^{-}|H_{o}\right] \end{array}\right)=\left(\begin{array}{cc} \frac{1}{\sqrt{2+2\varphi }} & \frac{1}{\sqrt{2+2\varphi }}\\ \frac{1}{\sqrt{2-2\varphi }} & \frac{-1}{\sqrt{2-2\varphi }} \end{array}\right)\left(\begin{array}{c} 0\\ 0 \end{array}\right)+\left(\begin{array}{c} 0\\ 0 \end{array}\right)\;\left(\begin{array}{c} E\left[NES_{rAB}^{+}|H_{o}\right]\\ E\left[NES_{rAB}^{-}|H_{o}\right] \end{array}\right)=\left(\begin{array}{c} 0\\ 0 \end{array}\right)$$
(18)$$\left(\begin{array}{cc} V𝕒𝕣\left[NES_{rAB}^{+}|H_{o}\right] & C𝕠𝕧\left[NES_{rAB}^{+},NES_{rAB}^{-}|H_{o}\right]\\ C𝕠𝕧\left[NES_{rAB}^{+},NES_{rAB}^{-}|H_{o}\right] & V𝕒𝕣\left[NES_{rAB}^{-}|H_{o}\right] \end{array}\right)=\ldots =\left(\begin{array}{cc} \frac{1}{\sqrt{2+2\varphi }} & \frac{1}{\sqrt{2+2\varphi }}\\ \frac{1}{\sqrt{2-2\varphi }} & \frac{-1}{\sqrt{2-2\varphi }} \end{array}\right)\left(\begin{array}{cc} 1 & \varphi \\ \varphi & 1 \end{array}\right)\left(\begin{array}{cc} \frac{1}{\sqrt{2+2\varphi }} & \frac{1}{\sqrt{2-2\varphi }}\\ \frac{1}{\sqrt{2+2\varphi }} & \frac{-1}{\sqrt{2-2\varphi }} \end{array}\right)\;\left(\begin{array}{cc} V𝕒𝕣\left[NES_{rAB}^{+}|H_{o}\right] & C𝕠𝕧\left[NES_{rAB}^{+},NES_{rAB}^{-}|H_{o}\right]\\ C𝕠𝕧\left[NES_{rAB}^{+},NES_{rAB}^{-}|H_{o}\right] & V𝕒𝕣\left[NES_{rAB}^{-}|H_{o}\right] \end{array}\right)=\ldots =\left(\begin{array}{cc} \frac{1}{\sqrt{2+2\varphi }}+\frac{\varphi }{\sqrt{2+2\varphi }} & \frac{\varphi }{\sqrt{2+2\varphi }}+\frac{1}{\sqrt{2+2\varphi }}\\ \frac{1}{\sqrt{2-2\varphi }}-\frac{\varphi }{\sqrt{2-2\varphi }} & \frac{\varphi }{\sqrt{2-2\varphi }}-\frac{1}{\sqrt{2-2\varphi }} \end{array}\right)\left(\begin{array}{cc} \frac{1}{\sqrt{2+2\varphi }} & \frac{1}{\sqrt{2-2\varphi }}\\ \frac{1}{\sqrt{2+2\varphi }} & \frac{-1}{\sqrt{2-2\varphi }} \end{array}\right)\;V𝕒𝕣\left[NES_{rAB}^{+}|H_{o}\right]=\left(\frac{1}{\sqrt{2+2\varphi }}\right)\left(\frac{1}{\sqrt{2+2\varphi }}+\frac{\varphi }{\sqrt{2+2\varphi }}\right)+\ldots +\left(\frac{1}{\sqrt{2+2\varphi }}\right)\left(\frac{\varphi }{\sqrt{2+2\varphi }}+\frac{1}{\sqrt{2+2\varphi }}\right)\;V𝕒𝕣\left[NES_{rAB}^{+}|H_{o}\right]=\left(\frac{1}{2+2\varphi }\right)+\left(\frac{\varphi }{2+2\varphi }\right)+\left(\frac{\varphi }{2+2\varphi }\right)+\left(\frac{1}{2+2\varphi }\right)\;V𝕒𝕣\left[NES_{rAB}^{+}|H_{o}\right]=\left(\frac{2+2\varphi }{2+2\varphi }\right)=1\;V𝕒𝕣\left[NES_{rAB}^{-}|H_{o}\right]=\left(\frac{1}{\sqrt{2-2\varphi }}\right)\left(\frac{1}{\sqrt{2-2\varphi }}-\frac{\varphi }{\sqrt{2-2\varphi }}\right)-\ldots -\left(\frac{1}{\sqrt{2-2\varphi }}\right)\left(\frac{\varphi }{\sqrt{2-2\varphi }}-\frac{1}{\sqrt{2-2\varphi }}\right)\;V𝕒𝕣\left[NES_{rAB}^{-}|H_{o}\right]=\left(\frac{1}{2-2\varphi }\right)-\left(\frac{\varphi }{2-2\varphi }\right)-\left(\frac{\varphi }{2-2\varphi }\right)+\left(\frac{1}{2-2\varphi }\right)\;V𝕒𝕣\left[NES_{rAB}^{-}|H_{o}\right]=\left(\frac{2-2\varphi }{2-2\varphi }\right)=1$$
(19)$$C𝕠𝕧\left[NES_{rAB}^{+},NES_{rAB}^{-}|H_{o}\right]=\ldots =\left(\frac{1}{\sqrt{2+2\varphi }}\right)\left(\frac{1}{\sqrt{2-2\varphi }}-\frac{\varphi }{\sqrt{2-2\varphi }}\right)+\left(\frac{1}{\sqrt{2+2\varphi }}\right)\left(\frac{\varphi }{\sqrt{2-2\varphi }}-\frac{1}{\sqrt{2-2\varphi }}\right)\;C𝕠𝕧\left[NES_{rAB}^{+},NES_{rAB}^{-}|H_{o}\right]=\ldots =\left(\frac{1}{\sqrt{2+2\varphi }\sqrt{2-2\varphi }}\right)-\left(\frac{\varphi }{\sqrt{2+2\varphi }\sqrt{2-2\varphi }}\right)+\left(\frac{\varphi }{\sqrt{2+2\varphi }\sqrt{2-2\varphi }}\right)-\ldots -\left(\frac{1}{\sqrt{2+2\varphi }\sqrt{2-2\varphi }}\right)\;C𝕠𝕧\left[NES_{rAB}^{+},NES_{rAB}^{-}|H_{o}\right]=\left(\frac{1-1+\varphi -\varphi }{\sqrt{2+2\varphi }\sqrt{2-2\varphi }}\right)=0$$

We find that $\left\{NES_{rAB}^{+},NES_{rAB}^{-}\right\}$ is a bivariate standard normal random vector under the null hypothesis of gene set analysis where the mean of each marginal is equal to zero, the variance of equal marginal is equal to one, and the covariance is equal to zero. Motivated by the previous discussion about the behavior of the positive Normalized Enrichment Score under the positive alternative hypothesis and the behavior of the negative Normalized Enrichment Score under the negative alternative hypothesis, we can calculate the statistical significance of each element of this vector using the standard normal cumulative distribution function as follows (Equation (20)):(20)$$p_{rAB}^{+}=1-\Phi \left(NES_{rAB}^{+}\right)\;p_{rAB}^{-}=\Phi \left(NES_{rAB}^{-}\right)$$

These *p*-values can be interpreted in a one-tailed manner as providing evidence against the null hypothesis in favor of the positive or negative version of the alternative hypothesis, respectively. Under the frequentist paradigm of null hypothesis significance testing, a sufficiently small ($p_{rAB}^{+}$) would lead us to reject the null hypothesis in favor of the positive alternative hypothesis, whereas a sufficiently small ($p_{rAB}^{-}$) would lead us to reject the null hypothesis in favor of the negative alternative hypothesis. Furthermore, through an appeal to the Neyman–Pearson lemma, we can motivate the use of the ratio between these two one-sided *p*-values, which is equivalent to a likelihood ratio, to decide on the most likely form of the alternative hypothesis in a manner that is uniformly most powerful.

In order to control the overall Type I error rate of NaRnEA, we recognize that selecting the most likely form of the alternative hypothesis based on the minimum of these one-tailed *p*-values constitutes a form of multiple hypothesis testing that must be corrected for in a manner that accounts for the dependence between these one-tailed *p*-values under the null hypothesis of gene set analysis. Since these one-tailed *p*-values are calculated from the positive Normalized Enrichment Score and the negative Normalized Enrichment Score, their statistical dependence under the null hypothesis of gene set analysis can be established using the following Markov Chain (Equation (21)):(21)$$p_{rAB}^{+}↔NES_{rAB}^{+}↔NES_{rAB}^{-}↔p_{rAB}^{-}$$

It follows from the Data Processing Inequality Theorem [19] that ($I\left[p_{rAB}^{+};p_{rAB}^{-}|H_{o}\right]\leq I\left[NES_{rAB}^{+};NES_{rAB}^{-}|H_{o}\right]$). Since the positive Normalized Enrichment Score and the negative Normalized Enrichment Score are jointly normally distributed under the null hypothesis of gene set analysis, we can calculate the mutual information between them using the following formula:$$I\left[NES_{rAB}^{+};NES_{rAB}^{-}|H_{o}\right]=\left(-\frac{1}{2}\right)\log\left(1-\rho {\left[NES_{rAB}^{+},NES_{rAB}^{-}|H_{o}\right]}^{2}\right)$$

We have already shown that, as a result of the affine transformation that produced the positive Normalized Enrichment Score and the negative Normalized Enrichment Score, the Pearson product moment correlation between these test statistics is equal to zero under the null hypothesis of gene set analysis. Thus, it immediately follows that the mutual information between the positive Normalized Enrichment Score and the negative Normalized Enrichment Score is equal to zero under the null hypothesis of gene set analysis. As a result, ($p_{rAB}^{+}$) and ($p_{rAB}^{-}$) are independent under the null hypothesis of gene set analysis. Thus, we can correct for our multiple hypothesis testing to obtain the final *p*-value for NaRnEA as follows (Equation (22)):(22)$$p_{rAB}=1-{\left(1-\min\left(p_{rAB}^{+},p_{rAB}^{-}\right)\right)}^{2}$$

We recognize that this is a two-sided *p*-value since it may be statistically significant under either the positive alternative hypothesis or the negative alternative hypothesis. We can use the magnitude of this final two-sided *p*-value and our knowledge of whether ($p_{rAB}^{+}$) or ($p_{rAB}^{-}$) is smaller to calculate the final Normalized Enrichment Score for the regulon gene set of the *r*th regulator in the nonparametric differential gene expression signature computed between the test phenotype (A) and the reference phenotype (B) for NaRnEA as follows (Equation (23)):(23)$$NES_{rAB}=\{\begin{array}{c} \Phi^{-1}\left(1-\frac{p_{rAB}}{2}\right)\;if\;p_{rAB}^{+}<p_{rAB}^{-}\;\\ \Phi^{-1}\left(\frac{p_{rAB}}{2}\right)\;if\;p_{rAB}^{+}>p_{rAB}^{-} \end{array}$$

By virtue of its construction, the NaRnEA Normalized Enrichment Score has the following properties:$p\left(NES_{rAB}|H_{o}\right)\overset{D}{\to }N\left(0,1\right)$The final Normalized Enrichment Score for the *r*th regulator is a standard normal random variable when the regulon gene set for the *r*th regulator is not enriched in the nonparametric differential gene expression signature computed between the test phenotype (A) and the reference phenotype (B). Formally, the rate of this asymptotic convergence depends on the Association Weight and Association Mode values for the gene set members in accordance with the Berry–Esseen Theorem for non-identically distributed summands.$E\left[NES_{rAB}|H_{a}^{+}\right]>0$The expected value of the final Normalized Enrichment Score for the *r*th regulator is positive when the regulon gene set for the *r*th regulator is positively enriched in the nonparametric differential gene expression signature computed between the test phenotype (A) and the reference phenotype (B).$E\left[NES_{rAB}|H_{a}^{-}\right]<0$The expected value of the final Normalized Enrichment Score for the *r*th regulator is negative when the regulon gene set for the *r*th regulator is negatively enriched in the nonparametric differential gene expression signature computed between the test phenotype (A) and the reference phenotype (B).

The asymptotic normality of the NaRnEA Normalized Enrichment Score under the null hypothesis of gene set analysis follows from the Lindeberg Central Limit Theorem, for which Lindeberg’s condition is sufficient. Since the Association Weight and Association Mode parameters for the members of the gene set are the reason that the summands are not necessarily identical, we require that these parameters do not exhibit extreme imbalance, which would violate Lindeberg’s condition; we show that a simple nonparametric procedure for parameterizing regulon gene sets that have been inferred from context-specific transcriptional regulatory networks produces sufficiently well-balanced gene sets that fulfill Lindeberg’s condition, thus allowing NaRnEA to maintain adequate control of the Type I error rate for gene set analysis. We also note that if NaRnEA is applied to gene sets in which all members have equal Association Weight and Association Mode values, such as literature-derived gene sets, the asymptotic normality of the NaRnEA Normalized Enrichment Score under the null hypothesis of gene set analysis is guaranteed by the classical Lindeberg–Lévy Central Limit Theorem [21].

The NaRnEA Normalized Enrichment Score is an optimal and robust test statistic for gene set analysis due its nonparametric integration of differential gene expression signature values, its nuanced ability to apply differential weighting to gene set members, its flexibility regarding uncertainty in the monotonicity of transcriptional regulation, and the derivation of its null model using the information theoretic Principle of Maximum Entropy. However, since the NaRnEA two-sided *p*-value, which may be analytically calculated from the NaRnEA Normalized Enrichment Score using the standard normal cumulative distribution function, does not measure the magnitude of gene set enrichment, we also provide an effect size for NaRnEA.

To derive an effect size for NaRnEA, we first consider the Wilcoxon signed-rank test, another nonparametric null hypothesis significance test that returns the *T*-statistic. Like the NaRnEA Normalized Enrichment Score, the *T*-statistic has a mean of (0) and is approximately normally distributed under the null hypothesis of the Wilcoxon signed-rank test. If the *T-*statistic is divided by its maximum possible value (i.e., the total sum of ranks), the resulting effect size is known as the rank–biserial correlation; it has a maximum value of (1) and minimum value of (−1). We can leverage a similar approach to calculate the NaRnEA Proportional Enrichment Score (PES), which serves as the effect size for the enrichment of the *r*th regulon in the nonparametric differential gene expression signature computed between the test phenotype (A) and the reference phenotype (B), as follows (Equation (24)):(24)$$PES_{rAB}=\{\begin{array}{c} \frac{NES_{rAB}}{\left|\max\left(NES_{rAB}\right)\right|}\;if\;NES_{rAB}>0\\ \frac{NES_{rAB}}{\left|\min\left(NES_{rAB}\right)\right|}\;if\;NES_{rAB}<0 \end{array}$$

By virtue of its construction, the NaRnEA Proportional Enrichment Score has the following properties:$PES_{rAB}\in \left[-1,\;1\right]$The Proportional Enrichment Score for the regulon gene set of the *r*th regulator is less than or equal to (1) and greater than or equal to (−1).$E\left[PES_{rAB}|H_{o}\right]=0$The expected value of the Proportional Enrichment Score for the regulon gene set of the *r*th regulator is equal to (0) when the regulon gene set of the *r*th regulator is not enriched in the nonparametric differential gene expression signature computed between the test phenotype (A) and the reference phenotype (B).$E\left[PES_{rAB}|H_{a}^{+}\right]>0$The expected value of the Proportional Enrichment Score for the regulon gene set of the *r*th regulator is positive when the regulon gene set of the *r*th regulator is positively enriched in the nonparametric differential gene expression signature computed between the test phenotype (A) and the reference phenotype (B).$E\left[PES_{rAB}|H_{a}^{-}\right]<0$The expected value of the Proportional Enrichment Score for the regulon gene set of the *r*th regulator is negative when the regulon gene set of the *r*th regulator is negatively enriched in the nonparametric differential gene expression signature computed between the test phenotype (A) and the reference phenotype (B).

The effect size for NaRnEA (i.e., Proportional Enrichment Score) is calculated from the test statistic for NaRnEA (i.e., Normalized Enrichment Score) in the same way that the effect size for the Wilcoxon signed-rank test (i.e., rank–biserial correlation) is calculated from the test statistic for the Wilcoxon signed-rank test (i.e., the *T*-statistic). Thus, the Proportional Enrichment Score can be interpreted as type of nonparametric correlation coefficient. A confidence interval for the NaRnEA Proportional Enrichment Score can be estimated by applying the Fisher z-transformation to achieve approximate normality of this effect size under the alternative hypothesis; the associated standard error of the Fisher z-transformed NaRnEA Proportional Enrichment Score can be estimated by applying a suitable resampling procedure such as bootstrapping members of the gene set [22].

If the two-sided *p*-value computed from the NaRnEA Normalized Enrichment Score is statistically significant, we can reject the null hypothesis of gene set analysis in favor of either the positive alternative hypothesis or the negative alternative hypothesis based on the sign of the Normalized Enrichment Score. In a manner inspired by Subramanian et al. [9], we can subsequently identify the members of the gene set that contribute most significantly to the enrichment by calculating a Leading Edge Score for each gene set member as follows (Equation (25)):(25)$$LES_{rgAB}=\{\begin{array}{c} \;\left(1-\left|AM_{rg}\right|\right)\left(r_{gAB}\right)+\left(AM_{rg}\right)\left(r_{gAB}s_{gAB}\right)\;\;if\;NES_{rAB}>0\\ \;\left(1-\left|AM_{rg}\right|\right)\left(r_{gAB}\right)-\left(AM_{rg}\right)\left(r_{gAB}s_{gAB}\right)\;\;if\;NES_{rAB}<0 \end{array}$$

If either the positive alternative hypothesis or the negative alternative hypothesis is true, we would expect the *g*th gene to have a strongly positive Leading Edge Score with respect to the *r*th regulator if the *g*th gene is contributing to the enrichment of the regulon gene set of the *r*th regulator in the nonparametric differential gene expression signature computed between the test phenotype (A) and the reference phenotype (B). Thus, we can calculate the statistical significance of this Leading Edge Score for the *g*th gene with respect to the *r*th regulator using the Maximum Entropy null model for gene set analysis as follows (Equation (26)):(26)$$p_{rgAB}=\{\begin{array}{c} \frac{1}{G}\overset{G}{\underset{k=1}{\sum }}I\left\{\left(1-\left|AM_{rg}\right|\right)\left(r_{kAB}\right)+\left(AM_{rg}\right)\left(r_{kAB}s_{kAB}\right)\geq LES_{rgAB}\right\}\;if\;NES_{rAB}>0\\ \frac{1}{G}\overset{G}{\underset{k=1}{\sum }}I\left\{\left(1-\left|AM_{rg}\right|\right)\left(r_{kAB}\right)-\left(AM_{rg}\right)\left(r_{kAB}s_{kAB}\right)\geq LES_{rgAB}\right\}\;if\;NES_{rAB}<0 \end{array}$$
where I{·} is the indicator function. These post hoc, one-tailed Leading Edge *p*-values may be adjusted for multiple hypothesis testing to identify those gene set members that contribute most significantly to the gene set enrichment. Importantly, the Leading Edge Score does not depend on the Association Weight of the *g*th gene with respect to the *r*th regulator; this ensures that the selection of genes that belong to the leading edge of the gene set a posteriori is not biased by any measure of gene set member importance that has been determined a priori.

### 2.2. The Algorithm for the Reconstruction of Accurate Cellular Networks 3 (ARACNe3)

ARACNe3 is an updated implementation of the Algorithm for the Reconstruction of Accurate Cellular Networks. The goal of ARACNe3 is to reverse-engineer a context-specific transcriptional regulatory network that consists of bivariate interactions between a set of predefined, putative transcriptional regulators and potential transcriptional targets.

ARACNe3 accepts properly normalized gene expression profiles that correspond to independent samples from a single biological phenotype. Like previous versions of the algorithm, ARACNe3 recommends that users reverse-engineer multiple estimates of the transcriptional regulatory network topology and integrate these to form a consensus network. Previously, ARACNe-AP recommended that the estimates of the transcriptional regulatory network topology should be reverse-engineered in a decorrelated manner by sampling from the original set of gene expression profiles with replacement (i.e., bootstrapping). While this approach is commonly employed in the field of ensemble machine learning (e.g., random forest bagging [23]), we find that sampling gene expression profiles with replacement increases the bias and variance of the adaptive partitioning mutual information (APMI) estimator; the increase in bias occurs when sampling with replacement produces regions in the joint probability distribution with higher density due to replicated data points, while the increase in variance occurs because these fluctuations in the joint probability distribution occur stochastically between different iterations of the bootstrapping procedure.

To avoid these pitfalls, ARACNe3 generates decorrelated individual networks by sampling ($1-\frac{1}{e}\approx 63.21\%$) of the gene expression profiles without replacement each time; this is equal to the probability of a unique sample appearing in a single bootstrap and thus achieves the same level of decorrelation between individual estimates of the context-specific transcriptional regulatory network as bootstrapping without unduly increasing the bias or variance of the APMI estimator. ARACNe3 estimates the null distribution for mutual information by applying the APMI estimator to ~1,000,000 pairs of shuffled, copula-transformed gene expression marginals (i.e., gene expression marginals are rank-transformed and divided by the number of gene expression profiles plus one to ensure the marginals are uniform). ARACNe3 then fits a piecewise null model to these null mutual information estimates where an empirical cumulative distribution function is used for the body of the null model up to the 95th percentile of the data and the tail of the null model is fit analytically using robust linear regression applied to logarithmically transformed tail probabilities past the 95th percentile with the mblm R package from CRAN [24]. The ARACNe3 piecewise null model controls the Type I error rate for the APMI estimator more accurately than the null model implemented in ARACNe-AP and allows ARACNe3 to perform the first round of individual network pruning based on the control of the False Discovery Rate (FDR), resulting in a substantial gain in power over previous versions of the algorithm that performed the first round of individual network pruning based on the control of the Family-Wise Error Rate (FWER).

ARACNe3 performs the second round of individual network pruning in a manner nearly identical to previous versions of the algorithm; briefly, all three-gene cliques that remain after the first round of individual network pruning are identified, and the weakest edge of each three-gene clique is removed from the network. The edges that remain after both rounds of pruning constitute an ARACNe3-inferred individual network. This procedure is carried out until one of two stopping criteria is met: (1) a prespecified maximum number of individual networks have been reverse-engineered, or (2) each putative transcriptional regulator has been assigned a prespecified minimum number of unique targets. The individual networks are then integrated to form an ARACNe3-inferred consensus transcriptional regulatory network. The mutual information and Spearman correlation for each putative transcriptional regulatory interaction in the ensemble network are estimated a final time using all gene expression profiles for greater accuracy.

The ARACNe3-inferred regulon gene set for the *r*th transcriptional regulator is constructed by extracting all putative transcriptional regulatory interactions for the *r*th transcriptional regulator from the ARACNe3-inferred consensus transcriptional regulatory network. The Association Weight values are calculated by sorting all putative target genes based on (1) the number of individual networks in which they appeared as targets of the *r*th transcriptional regulator and (2) the final estimated mutual information between the *r*th transcriptional regulator and target gene. A copula transformation is then applied to the Association Weight values to ensure that the ARACNe3-inferred regulon gene sets are sufficiently well balanced to meet Lindeberg’s condition and guarantee the asymptotic standard normality of the NaRnEA Normalized Enrichment Score under the null hypothesis of gene set analysis. The Association Mode values are taken to be the Spearman correlation coefficient between the *r*th transcriptional regulator and each regulon gene set member as estimated from all gene expression profiles.

The lung adenocarcinoma (LUAD) context-specific transcriptional regulatory network was reverse-engineered with ARACNe3 from 476 unpaired primary tumor gene expression profiles from TCGA using 2491 putative transcriptional regulators. Gene expression profiles were downloaded using the TCGAbiolinks R package from Bioconductor [25] and normalized for sequencing depth prior to network reverse engineering (i.e., counts per million). The first round of individual network pruning was carried out with a threshold for mutual information calculated to control the FDR at 5%. Individual networks were reverse-engineered until each putative transcriptional regulator had at least 50 unique transcriptional targets; this was achieved after seven iterations. The final consensus ARACNe3-inferred transcriptional regulatory network for LUAD consists of 2491 regulators, 19,350 targets, and 790,200 regulatory interactions.

The colon adenocarcinoma (COAD) context-specific transcriptional regulatory network was reverse-engineered with ARACNe3 from 437 unpaired primary tumor gene expression profiles from TCGA using 2491 putative transcriptional regulators. Gene expression profiles were downloaded using the TCGAbiolinks R package from Bioconductor and normalized for sequencing depth prior to network reverse engineering (i.e., counts per million). The first round of individual network pruning was carried out with a threshold for mutual information calculated to control the FDR at 5%. Individual networks were reverse-engineered until each putative transcriptional regulator had at least 50 unique transcriptional targets; this was achieved after seven iterations. The final consensus ARACNe3-inferred transcriptional regulatory network for COAD consists of 2491 regulators, 19,350 targets, and 675,373 regulatory interactions.

The head and neck squamous cell carcinoma (HNSC) context-specific transcriptional regulatory network was reverse-engineered with ARACNe3 from 457 unpaired primary tumor gene expression profiles from TCGA using 2491 putative transcriptional regulators. Gene expression profiles were downloaded using the TCGAbiolinks R package from Bioconductor and normalized for sequencing depth prior to network reverse engineering (i.e., counts per million). The first round of individual network pruning was carried out with a threshold for mutual information calculated to control the FDR at 5%. Individual networks were reverse-engineered until each putative transcriptional regulator had at least 50 unique putative transcriptional targets; this was achieved after 12 iterations. The final consensus ARACNe3-inferred transcriptional regulatory network for HNSC consists of 2491 regulators, 19,350 targets, and 812,199 regulatory interactions.

### 2.3. Gene Set Enrichment Analysis (GSEA)

GSEA accepts properly normalized gene expression profiles from samples representing a test phenotype and a reference phenotype; the differential expression of each gene is then estimated using the Signal-to-Noise Ratio (SNR). The enrichment of a gene set in this gene expression signature is calculated with a weighted Kolmogorov–Smirnov-like statistic (i.e., the GSEA enrichment score). Subramanian et al. [9] recommended that the null distribution of the GSEA enrichment score for a particular gene set should be approximated using an empirical phenotype-based permutation procedure. Alternatively, if there are not enough samples to generate the number of desired phenotype-based permutations, an empirical gene-based permutation procedure may be used to approximate the null distribution of the GSEA enrichment score.

Paired gene expression profiles from primary tumors and phenotype-matched normal tissue samples from TCGA were normalized using a blinded DESeq2 [26] variance-stabilizing transformation prior to analysis with GSEA, which was performed as described previously by Subramanian et al. [9] using the Java command line implementation of GSEA from the Molecular Signatures Database (http://www.gsea-msigdb.org/gsea/downloads.jsp (accessed on 1 October 2020)). The GSEA null model was estimated using 1000 sample-shuffling permutations. Empirical two-sided *p*-values returned by GSEA were corrected to a minimum of (1/1001), the smallest possible two-sided *p*-value for an empirical null model constructed from 1000 sample-shuffling permutations [27].

### 2.4. Analytical-Rank-Based Enrichment Analysis (aREA)

aREA accepts properly normalized gene expression profiles from samples representing a test phenotype and a reference phenotype; the differential expression of each gene is then estimated using Welch’s unpaired *t*-test [28]. The enrichment of a gene set in the resulting differential gene expression signature is calculated using a three-tailed approach, returning the aREA enrichment score test statistic. Alvarez et al. [4] recommended that the null distribution of the aREA enrichment score for a particular gene set should also be approximated using an empirical phenotype-based permutation procedure. Alternatively, if there are not enough samples to generate the number of desired phenotype-based permutations, an analytical approach may be used to approximate the null distribution of the aREA enrichment score.

Paired gene expression profiles from primary tumors and phenotype-matched normal tissue samples from TCGA were normalized using a blinded DESeq2 variance-stabilizing transformation prior to analysis with aREA. The VIPER R package from Bioconductor was used to run aREA, as described previously by Alvarez et al. [4]. The aREA empirical null model was estimated using 1000 sample-shuffling permutations. Empirical two-sided *p*-values returned by aREA were corrected to a minimum of (1/1001), the smallest possible two-sided *p*-value for an empirical null model constructed from 1000 sample-shuffling permutations.

### 2.5. Clinical Proteomic Tumor Analysis Consortium (CPTAC) Differential Protein Abundance

Log-ratio normalized protein abundance data for primary tumors and phenotype-matched normal tissue samples were downloaded from CPTAC for the LUAD, COAD, and HNSC cancer types (http://linkedomics.org (accessed on 1 October 2020)) [29]. Data were loaded into R, and the differential abundance of each protein between primary tumors and phenotype-matched normal tissue was estimated with a two-tailed Mann–Whitney U test [30]. Gene name conversion was performed using the biomaRt R package from Bioconductor [31].

### 2.6. Plotting and Visualization

All figures were created in R using the ggplot2 R package from CRAN [32].

### 2.7. Statistical Analysis

*p*-values were corrected for multiple hypothesis testing to control the FDR according to the methodology of Benjamini and Hochberg or to control the FWER according to the methodology of Bonferroni [33]. The 95% confidence intervals for the binomial test of proportions were computed using the procedure of Clopper and Pearson [34].

## 3. Results

### 3.1. Evaluating the Sensitivity and Specificity of NaRnEA for Gene Set Analysis

We evaluated NaRnEA by performing gene set analysis using gene expression data from The Cancer Genome Atlas (TCGA) for lung adenocarcinoma (LUAD), colon adenocarcinoma (COAD), and head and neck squamous cell carcinoma (HNSC); these cancer types were selected because of (1) the availability of phenotype-matched primary tumor and normal tissue RNA-Seq gene expression profiles in TCGA and (2) the availability of phenotype-matched primary tumor and normal tissue mass spectrometry protein abundance profiles in the Clinical Proteomic Tumor Analysis Consortium (CPTAC). Crucially, the differential protein abundance inferred from mass spectrometry data in CPTAC provides orthogonal validation for the differential protein activity inferred from gene expression data in TCGA. For each TCGA cohort, we separated the primary tumor gene expression profiles into two groups based on whether each primary tumor was submitted with or without an associated adjacent normal tissue sample; we refer to these as paired and unpaired primary tumor gene expression profiles, respectively. From the unpaired primary tumor gene expression profiles, we reverse-engineered a context-specific transcriptional regulatory network for each cancer type with ARACNe3 for 2491 putative transcriptional regulatory proteins (i.e., transcription factors, co-transcription factors, epigenetic modifying enzymes) [4]. From each context-specific ARACNe3-inferred transcriptional regulatory network, we extracted all edges associated with each regulator, producing the tumor-specific (TS) regulon gene sets.

Subsequently, we created an identical number of null model (NM) regulons by swapping out the members of each TS regulon with an equal number of genes selected at random from the complement of the corresponding TS regulon gene set. By virtue of their construction, the NM regulons are biologically meaningless and therefore will not be enriched in any differential gene expression signature. We used the gene expression profiles from 57 LUAD, 41 COAD, and 43 HNSC patient-matched primary tumor (i.e., test phenotype) and adjacent normal tissue (i.e., reference phenotype) samples from TCGA to estimate cohort-specific differential gene expression signatures. We then used NaRnEA to test for the enrichment of the NM regulons in the corresponding differential gene expression signatures. The number of statistically significantly enriched NM regulons was determined from the NaRnEA two-sided *p*-values based on the control of the False-Positive Rate (FPR < 0.05), False Discovery Rate (FDR < 0.05), or Family-Wise Error Rate (FWER < 0.05).

This analysis demonstrates that after correcting for multiple hypothesis testing, NaRnEA did not find any of the NM regulons to be statistically significantly enriched in the differential gene expression signatures computed between primary tumor and adjacent normal tissue samples from TCGA (Table 1). Crucially, this finding is demonstrated using biologically meaningless gene sets (i.e., NM regulons) and biologically meaningful differential gene expression signatures that should not alter the complex higher-order dependencies that Mootha et al. [8], Subramanian et al. [9], and Tamayo et al. [10] claim exist between gene set members under the null hypothesis of gene set analysis. This finding that NaRnEA maintains specificity while using a null model for gene set analysis that explicitly assumes that gene set members are independent when the gene set is not enriched in the differential gene expression signature effectively falsifies the primary claim underlying the validity of the empirical phenotype-based permutation null model used by both GSEA and aREA.

Having established the specificity of NaRnEA using the NM regulons, we subsequently evaluated the sensitivity of NaRnEA using the TS regulons. Given the substantial differences in gene expression between malignant and benign phenotypes, we expected that at least some fraction of the 2491 transcriptional regulatory proteins to which these TS regulons correspond would exhibit differential activity between the primary tumor and adjacent normal tissue samples from TCGA. However, we did not know a priori which subset of transcriptional regulatory proteins would exhibit differential activity, since no experimental methodology exists to measure the activity of transcriptional regulatory proteins in a systematic, high-throughput manner in vivo. Instead, we used NaRnEA to test for the enrichment of the TS regulons in the corresponding differential gene expression signatures computed between the primary tumor and adjacent normal tissue samples from TCGA; then, we used independent mass spectrometry data from CPTAC to determine whether the differential activity of the transcriptional regulatory proteins, inferred from TCGA gene expression data using NaRnEA, was correlated with the differential abundance of the transcriptional regulatory proteins, inferred from CPTAC proteomic data using a Mann–Whitney U (MWU) test.

The statistical dependence between differential protein activity inferred by NaRnEA from gene expression data in TCGA and differential protein abundance inferred by an MWU test from mass spectrometry data in CPTAC can be expressed by the following Markov Chain (Equation (27)):(27)$$NES_{rAB}\leftarrow DRA_{rAB}\leftarrow DPA_{rAB}\to MWU_{rAB}$$
where ($NES_{rAB}$) is the NaRnEA Normalized Enrichment Score of the *r*th regulator between the test phenotype (A) and the reference phenotype (B), ($DRA_{rAB}$) is the differential regulatory activity of the *r*th regulator between the test phenotype (A) and the reference phenotype (B), ($DPA_{rAB}$) is the differential protein abundance of the *r*th regulator between the test phenotype (A) and the reference phenotype (B), and ($MWU_{rAB}$) is the result of the MWU test inferring the differential protein abundance for the *r*th regulator between the test phenotype (A) and the reference phenotype (B). It follows from the Data Processing Inequality Theorem [19] that the mutual information between the NaRnEA-inferred differential protein activity from TCGA and the MWU-inferred differential protein abundance from CPTAC is a lower bound on the mutual information between the NaRnEA-inferred differential protein activity and the true differential protein activity; we expect this mutual information to be reduced by several factors, both technical and biological. Since the activity of a transcriptional regulatory protein depends on numerous post-translational events (e.g., nuclear localization, post-translational modification, cofactor binding, chromatin accessibility), the activity of the regulator and the abundance of the regulator will differ, thereby weakening the statistical dependency between random variables in the Markov Chain. From a technical perspective, the extent to which the MWU-inferred differential protein abundance from CPTAC agrees with the true differential protein abundance will depend strongly on the accuracy of the mass spectrometry experimental analysis and the accuracy of the MWU test. Similarly, the extent to which the NaRnEA-inferred differential protein activity from TCGA agrees with the true differential protein activity will depend on the accuracy of the gene expression profiling, the accuracy of the ARACNe3-inferred TS regulons, and the accuracy of NaRnEA as a statistical method. Taking into consideration this myriad of confounders, any statistically significant correlation between NaRnEA-inferred differential protein activity from TCGA- and MWU-inferred differential protein abundance from CPTAC offers strong support for the sensitivity and biological validity of NaRnEA as a gene set analysis method.

The number of TS regulons that were statistically significantly enriched in the differential gene expression signatures computed between primary tumor and adjacent normal tissue samples in each of the cancer types from TCGA was determined from the NaRnEA two-sided *p*-values based on the control of the False-Positive Rate (FPR < 0.05), False Discovery Rate (FDR < 0.05), or Family-Wise Error Rate (FWER < 0.05).

This analysis demonstrates that after correcting for multiple hypothesis testing, many of the TS regulons were inferred by NaRnEA to be statistically significantly enriched in these differential gene expression signatures from TCGA (Table 2). To determine whether this NaRnEA-inferred differential protein activity from TCGA agreed with MWU-inferred differential protein abundance from CPTAC, we compared primary tumor (n_tumor_ = 110, 97, 109) and phenotype-matched normal tissue (n_tissue_ = 101, 100, 64) samples in the LUAD, COAD, and HNSC cohorts from CPTAC, respectively. For each cancer type, we restricted our analysis to the transcriptional regulatory proteins for which a TS regulon and mass spectrometry data were available. We corrected the MWU test two-sided *p*-values from CPTAC for multiple hypothesis testing and classified each transcriptional regulatory protein as upregulated (UP, MWU test rank–biserial correlation > 0, FDR < 0.05), downregulated (DOWN, MWU test rank–biserial correlation < 0, FDR < 0.05), or not statistically significantly differentially abundant (NS, FDR ≥ 0.05). Similarly, we also corrected the NaRnEA two-sided *p*-values from TCGA for multiple hypothesis testing and classified each transcriptional regulatory protein as activated (UP, NaRnEA PES > 0, FDR < 0.05), deactivated (DOWN, NaRnEA PES < 0, FDR < 0.05), or not statistically significantly differentially activated (NS, FDR ≥ 0.05). We tested for an association between NaRnEA-inferred differential protein activity from TCGA- and MWU-inferred differential protein abundance from CPTAC using a three-by-three contingency table for LUAD (Table 3), COAD (Table 4), and HNSC (Table 5). The agreement between the rows and columns was evaluated using Kendall’s Tau-B correlation coefficient, which adjusts for tied values within each of the three marginal categories; the statistical significance of this dependence was calculated with a Chi-Squared Test.

This analysis revealed that the NaRnEA-inferred differential protein activity from TCGA was statistically significantly positively correlated with the MWU-inferred differential protein abundance from CPTAC for the LUAD (Kendall’s Tau-B = 0.3832, *p* = 7.481 × 10^−51^), COAD (Kendall’s Tau-B = 0.2913, *p* = 1.456 × 10^−18^), and HNSC (Kendall’s Tau-B = 0.3455, *p* = 5.237 × 10^−38^) cancer types. These results offer biological validity for the NaRnEA-inferred differential protein activity, further reinforcing that NaRnEA is a highly sensitive gene set analysis method. The NaRnEA-inferred differential protein activity for all putative transcriptional regulatory proteins can be visualized *en masse* by plotting the absolute value of the Normalized Enrichment Score vs. the Proportional Enrichment Score for each TS regulon using a modified version of a volcano plot [35] (Figure 1, Figure 2 and Figure 3). Alternatively, one can directly visualize the distribution of ARACNe3-inferred transcriptional regulatory targets in the nonparametric differential gene expression signature computed between primary tumor and adjacent normal tissue samples for a subset of the most differentially activated proteins using a Master Regulator Analysis plot (Figure 4, Figure 5 and Figure 6).

To demonstrate the utility of the NaRnEA Leading Edge analysis, we calculated the post hoc Leading Edge *p*-values for each of the TS regulons that were statistically significantly enriched in the differential gene expression signatures computed between the primary tumor and adjacent normal tissue samples from TCGA (FWER < 0.05). Since there was no overlap between the gene expression profiles that were used by ARACNe3 to reverse-engineer the context-specific transcriptional regulatory networks and the gene expression profiles that were used to estimate the differential gene expression signatures for these cancer types, the NaRnEA Leading Edge *p*-values and the ARACNe3-inferred Association Weight values would be independent under the null hypothesis that the NaRnEA Leading Edge analysis cannot identify the gene set members that contribute most significantly to the gene set enrichment. In support of the NaRnEA Leading Edge analysis, this analysis revealed that the ARACNe3-inferred Association Weight values and the NaRnEA-inferred post hoc Leading Edge *p*-values exhibited a statistically significantly negative Spearman correlation (FWER < 0.05) for the vast majority of TS regulons in TCGA LUAD (91.75%), TCGA COAD (90.56%), and TCGA HNSC (88.33%). This analysis was restricted to those TS regulons that NaRnEA inferred were statistically significantly enriched in the differential gene expression signatures computed between the corresponding primary tumor and adjacent normal tissue samples from TCGA (FWER < 0.05) since post hoc Leading Edge analysis is only interpretable for these gene sets.

### 3.2. Identifying Systematic Biases in Phenotype-Based Permutation Null Models for Gene Set Enrichment

To compare NaRnEA with GSEA and aREA, we first applied these alternative gene set analysis methods to test for the enrichment of the NM regulons in the gene expression data from TCGA for LUAD, COAD, and HNSC. The number of statistically significantly enriched NM regulons was determined for GSEA (Table 6) and aREA (Table 7) from the resulting two-sided *p*-values; statistical significance was established based on the control of the False-Positive Rate (FPR < 0.05), False Discovery Rate (FDR < 0.05), or Family-Wise Error Rate (FWER < 0.05).

This analysis demonstrates that, after correcting for multiple hypothesis testing, neither GSEA nor aREA found any of the NM regulons to be statistically significantly enriched in the differential gene expression signatures computed between primary tumor and adjacent normal tissue samples from TCGA; thus, we conclude that both methods adequately control the Type I error rate of gene set analysis. We subsequently applied GSEA and aREA to test for the enrichment of the TS regulons in the gene expression data from TCGA for LUAD, COAD, and HNSC. The number of statistically significantly enriched TS regulons was determined for GSEA (Table 8) and aREA (Table 9) from the resulting two-sided *p*-values; statistical significance was established based on the control of the False-Positive Rate (FPR < 0.05), False Discovery Rate (FDR < 0.05), or Family-Wise Error Rate (FWER < 0.05).

This analysis demonstrates that, after correcting for multiple hypothesis testing, neither GSEA nor aREA found any of the TS regulons to be statistically significantly enriched in the differential gene expression signatures computed between primary tumor and adjacent normal tissue samples from TCGA. Given that the NaRnEA-inferred differential protein activity from TCGA was significantly correlated with MWU-inferred differential protein abundance from CPTAC, we conclude that NaRnEA is significantly more sensitive than both GSEA and aREA; furthermore, the fact that NaRnEA did not identify any statistically significantly enriched NM regulons after correcting for multiple hypothesis testing demonstrates that this gain in sensitivity is achieved without loss of specificity.

The low sensitivity of GSEA and aREA, as evidenced by their inability to identify statistically significantly enriched TS regulons after correcting for multiple hypothesis testing, can be attributed directly to their reliance on the empirical phenotype-based permutation null model, which we have shown to be unnecessary for achieving adequate control of the gene set analysis Type I error rate. In order to determine why the use of the empirical phenotype-based permutation null model decreases the sensitivity of these methods, we applied this procedure to the gene expression data from TCGA while determining how many primary tumor and adjacent normal tissue samples were distributed to the null test phenotype and null reference phenotype during each permutation. Then, for each of the sample-shuffling permutations, we estimated a null differential gene expression signature between the null test phenotype samples and the null reference phenotype samples using an MWU test; we repeated this process 1000 times for each cancer type. We then tested for the enrichment of the ARACNe3-inferred TS regulons in each of these null differential gene expression signatures with NaRnEA.

After correcting the resulting NaRnEA two-sided *p*-values for multiple hypothesis testing (FWER < 0.05), we found that some of the TS regulons were statistically significantly enriched in each of the null differential gene expression signatures computed from TCGA LUAD (minimum = 112, median = 568, maximum = 1053), TCGA COAD (minimum = 65, median = 532, maximum = 1041), and TCGA HNSC (minimum = 78, median = 489, maximum = 916). Since we established that NaRnEA adequately controls the Type I error rate of gene set analysis, we can conclude that the enrichment of the TS regulons was not an artifact; rather, this finding suggests that each of the null differential gene expression signatures exhibited some degree of correlation with the original differential gene expression signature.

To test for this, we calculated the Spearman correlation between each null differential gene expression signature and the original differential gene expression signature. After correcting the two-sided *p*-values for multiple hypothesis testing (FWER < 0.05), we found that nearly all of the null differential gene expression signatures from TCGA LUAD (94.7%), TCGA COAD (95%), and TCGA HNSC (93.2%) were statistically significantly correlated with the original differential gene expression signature from the same cancer type. Furthermore, we found that this correlation between each null differential gene expression signature and the original differential gene expression signature was strongly associated with the degree of imbalance between the corresponding null phenotypes; for example, when the null test phenotype contained a greater number of primary tumor samples than adjacent normal tissue samples, the resulting null differential gene expression signature was more likely to be positively correlated with the original differential gene expression signature (Figure 7, Figure 8 and Figure 9).

This analysis provides an immediate explanation for the reduced sensitivity of GSEA and aREA: the empirical phenotype-based permutation null models leveraged by each of these methods are contaminated with enrichment test statistics that do not strictly follow the null distribution due to the enrichment of the gene sets in a portion of the null differential gene expression signatures. Thus, in addition to demonstrating that NaRnEA adequately controls the Type I error rate of gene set analysis with an analytical null model derived using the information theoretic Principle of Maximum Entropy, we have also shown that systematic biases in the empirical phenotype-based permutation null model leveraged by GSEA and aREA can fully explain the substantial difference in sensitivity between these methods.

### 3.3. Examining the Alternative Null Model for GSEA

Having established that the empirical phenotype-based permutation null model for gene set analysis is both systematically flawed and unnecessary to maintain adequate control of the Type I error rate, we next examined the alternative null model for GSEA that Subramanian et al. [9] described as follows:
*“… [GSEA] can be applied to ranked gene lists arising in other settings. Genes may be ranked based on the differences seen in a small data set, with too few samples to allow rigorous evaluation of significance levels by permuting the class labels. In these cases, a P value can be estimated by permuting the genes, with the result that genes are randomly assigned to the sets while maintaining their size. This approach is not strictly accurate: because it ignores gene-gene correlations, it will overestimate the significance levels and may lead to false positives. Nonetheless, it can be useful for hypothesis generation”.*

Tamayo et al. [10] formally describe the GSEA enrichment score as the following weighted variation of the two-sample Kolmogorov–Smirnov statistic (Equation (28)):(28)$$S_{k}^{GSEA}=sup_{i=1,\ldots ,N}\left(F_{i}^{g_{k}}-F_{i}^{{\bar{g}}_{k}}\right)\;F_{i}^{g_{k}}=\frac{\sum_{h=1}^{i}{\left|s_{h}\right|}^{\alpha }I_{h}}{\sum_{h=1}^{N}{\left|s_{h}\right|}^{\alpha }I_{h}}\;F_{i}^{{\bar{g}}_{k}}=\frac{\sum_{h=1}^{i}\left(1-I_{h}\right)}{\left(N-n_{k}\right)}\;I_{h}=\{\begin{array}{c} 1\;if\;h\in g_{k}\\ 0\;if\;h\in {\bar{g}}_{k} \end{array}$$
where ($S_{k}^{GSEA}$) is the GSEA enrichment score test statistic for the *k*th gene set, (sup) is the supremum operator, ($F_{i}^{g_{k}}$) is the component of the running sum statistic computed at the *i*th ranked gene in the gene expression signature that corresponds with the gene set, ($F_{i}^{{\bar{g}}_{k}}$) is the component of the running sum statistic computed at the *i*th ranked gene in the gene expression signature that corresponds with the gene set’s complement, ($I_{h}$) is an indicator variable that identifies whether the *h*th ranked gene belongs to the gene set or the gene set’s complement, ($s_{h}$) is the differential gene expression signature value of the *h*th ranked gene, (N) is the total number of genes in the differential gene expression signature, and ($n_{k}$) is the number of genes in the *k*th gene set. The exponential (α) determines the extent to which GSEA is sensitive to the magnitude of differential gene expression for each gene in the gene set when computing the enrichment score test statistic. Mootha et al. [8] originally recommended setting (α) to zero, which renders the GSEA enrichment score equivalent to the two-sample Kolmogorov–Smirnov test statistic, while Subramanian et al. [9] recommended setting (α) to one; they set (α) to one for GSEA based on the following observation:
*“In the original implementation, the running-sum statistic used equal weights at every step, which yielded high scores for sets clustered near the middle of the ranked list … These sets do not represent biologically relevant correlation with the phenotype. We addressed this issue by weighting the steps according to each gene’s correlation with a phenotype”.*

The behavior of GSEA, as implemented by Mootha et al. [8] with (α) set to zero, follows directly from the statistical formulation of the two-sample Kolmogorov–Smirnov test, which was created to test the null hypothesis that two sets of observations are independently and identically distributed from the same sampling distribution. By virtue of its design, the two-sample Kolmogorov–Smirnov test is sensitive to any sufficiently large deviation between the two sample-specific empirical cumulative distribution functions regardless of where in the set of observations that deviation occurs. Thus, the alternative hypothesis for the two-sample Kolmogorov–Smirnov test is far too broad for gene set analysis if one is interested in rejecting the null hypothesis only when the gene set members are enriched at the extremes of the differential gene expression signature. Subramanian et al. [9] attempted to modify GSEA by setting (α) equal to one; however, we can demonstrate that this pathological behavior is still present when the alternative empirical gene-based permutation null model for GSEA is used.

We constructed a new group of gene sets, which we refer to as totally null (TN) regulons, by replacing the members of each TS regulon with genes drawn at random from the set of genes for which the MWU differential gene expression two-sided *p*-value was greater than 0.50. Thus, these TN regulons consisted solely of genes that were not enriched at the extremes of the corresponding differential gene expression signature; therefore, an accurate gene set analysis method should not identify any statistically significant enrichment for these gene sets. We used NaRnEA (Table 10) and GSEA (Table 11) to test for the enrichment of the TN regulons in the corresponding differential gene expression signatures computed between the primary tumor and adjacent normal tissue samples from TCGA; here, we used the alternative empirical gene-based permutation null model for GSEA. The number of statistically significantly enriched TN regulons was determined from the resulting two-sided *p*-values; statistical significance was established based on the control of the False-Positive Rate (FPR < 0.05), False Discovery Rate (FDR < 0.05), or Family-Wise Error Rate (FWER < 0.05).

We determined that, after correcting for multiple hypothesis testing to control the False Discovery Rate, GSEA found 100% of the TN regulons to be statistically significantly enriched in the differential gene expression signatures computed between primary tumor and adjacent normal tissue samples from TCGA. In contrast, even without correcting for multiple hypothesis testing, NaRnEA did not find any of the TN regulons to be statistically significantly enriched in the differential gene expression signatures. These results falsify the secondary claim made by Subramanian et al. [9] and Tamayo et al. [10] that the weighted two-sample Kolmogorov–Smirnov test statistic prevents GSEA from detecting statistically significant enrichment for gene sets whose members exhibit biologically meaningless non-uniform distribution in the differential gene expression signature. Taken together, these findings demonstrate that GSEA exhibits significant and irreparable flaws that render its use as a gene set analysis method inappropriate regardless of whether the original empirical phenotype-based permutation null model or the alternative empirical gene-based permutation null model is employed.

### 3.4. Examining the Alternative Null Model for aREA

We also examined the alternative null model for aREA, which is described by Alvarez et al. [4] as follows:
*“… the statistical significance for the enrichment score is estimated by comparison to a null model generated by permuting the samples uniformly at random or by an analytic approach equivalent to shuffle the genes in the signatures uniformly at random … Gene shuffling can be approximated analytically as follows: according to the central limit theorem, the mean of a sufficiently large number of independent random variables will be approximately normally distributed. The enrichment score of our null hypothesis fulfill this condition, and we ensure a mean of zero and variance equal to one for the enrichment score under the null hypothesis by applying a quantile transformation based on the normal distribution to the rank-transformed gene expression signature before computing the enrichment score”.*

Alvarez et al. [4] invoked the Central Limit Theorem to claim that the aREA test statistic would be normally distributed with a mean of zero and a variance of one under the null hypothesis of gene set analysis. We directly tested this claim by permuting the values of the original differential gene expression signature to create 1000 shuffled differential gene expression signatures for each cancer type from TCGA. By virtue of this shuffling procedure, none of the TS regulons would be enriched in these shuffled differential gene expression signatures. In order to test the claim made by Alvarez et al. [4] that the aREA test statistics would be normally distributed with a mean of zero and a variance of one under the null hypothesis of gene set analysis, we used aREA to test for the enrichment of each TS regulon in each shuffled differential gene expression signature, producing 1000 aREA test statistics for each TS regulon in each cancer type from TCGA. We then tested the null hypothesis that the aREA test statistics for each TS regulon followed a standard normal distribution as Alvarez et al. [4] claim using a one-sample Kolmogorov–Smirnov test.

We found that, after correcting the resulting *p*-values for multiple hypothesis testing (FWER < 0.05), we rejected the null hypothesis that the aREA test statistics were normally distributed with a mean of zero and a variance of one under the null hypothesis of gene set analysis for 100% of the TS regulons in TCGA LUAD, TCGA COAD, and TCGA HNSC. This analysis demonstrates that the alternative null model provided for aREA does not behave in the manner described by Alvarez et al. [4]; unfortunately, no formal analysis could be conducted to identify the source of this discrepancy as the alternative null model for aREA was published without accompanying proof to provide its mathematical justification. Thus, we conclude that aREA also exhibits significant flaws as a gene set analysis method regardless of whether the original empirical phenotype-based permutation null model or the alternative analytical null model is employed. When we repeat this analysis with NaRnEA instead of aREA, we do not reject the null hypothesis for any of the TS regulons after correcting for multiple hypothesis testing (FWER < 0.05). Thus, while these results effectively falsify the alternative analytical null model for aREA, they demonstrate that the NaRnEA analytical null model behaves precisely as expected.

## 4. Discussion

It is widely appreciated that the rigor and reproducibility of scientific research depends on the use of computational and experimental methods that are sufficiently sensitive to make meaningful inferences while maintaining adequate control of the Type I error rate to reduce spurious findings [36]. Gene set analysis methods are being increasingly applied for hypothesis generation [37], precision oncology [15], systems pharmacology [38], analysis of single-cell transcriptomic data [39,40], and biomarker development [41]. Here, we demonstrate that NaRnEA significantly outperforms both GSEA and aREA for the purpose of gene set analysis in three independent cancer types; despite the widespread use of both competing methods, NaRnEA is the only method capable of consistently distinguishing between biologically coherent gene sets and gene sets constructed at random in these cohorts. Furthermore, the substantial agreement between NaRnEA-inferred differential protein activity in TCGA cohorts and MWU-inferred differential protein abundance in phenotype-matched CPTAC cohorts confirms that gene set enrichment inferred by NaRnEA cannot be explained away as erroneous false positives. Indeed, the specificity of NaRnEA is established by the fact that NaRnEA did not infer statistically significant enrichment for any of the NM regulons between primary tumor and adjacent normal tissue samples in TCGA after correcting for multiple hypothesis testing. We find that the low sensitivity of both GSEA and aREA can be attributed to their reliance on an empirical phenotype-based permutation null model that we show to be overwhelmingly confounded by genuine gene set enrichment due to persistent associations between null gene expression signatures and the original gene expression signatures. Finally, we show that the alternative null models for GSEA and aREA are statistically invalid due to similarity with the two-sample Kolmogorov–Smirnov test and an inaccurate mathematical framework, respectively. Future work will aim to demonstrate the application of NaRnEA to a wider range of malignant and non-malignant phenotypes of interest where either orthogonal data are available for validation (i.e., CPTAC protein abundance or similar) or in vitro follow-up experiments can validate potentially novel findings. Additionally, we aim to adapt this algorithm for the analysis of single-cell gene expression profiles. To encourage immediate use by members of the scientific community, both NaRnEA and ARACNe3 are freely available for research use on GitHub (https://github.com/califano-lab/NaRnEA (accessed on 3 March 2023)).

## Figures and Tables

**Figure 1 entropy-25-00542-f001:**
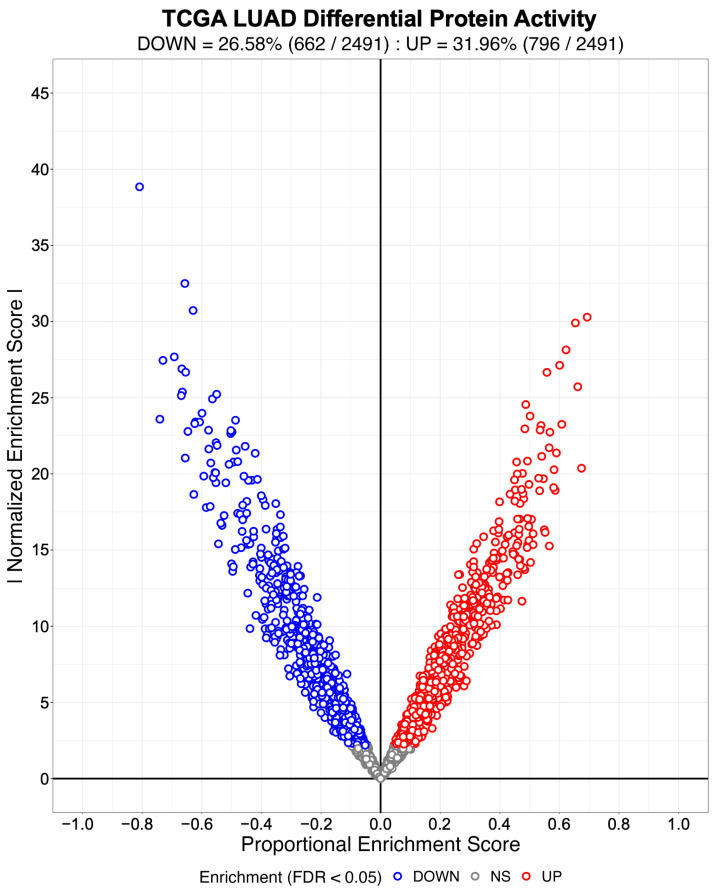
Differential protein activity volcano plot for the NaRnEA-inferred enrichment of TS-regulons in the differential gene expression signatures computed between primary tumors and phenotype-matched normal tissue samples from The Cancer Genome Atlas (TCGA) cohort for lung adenocarcinoma (LUAD).

**Figure 2 entropy-25-00542-f002:**
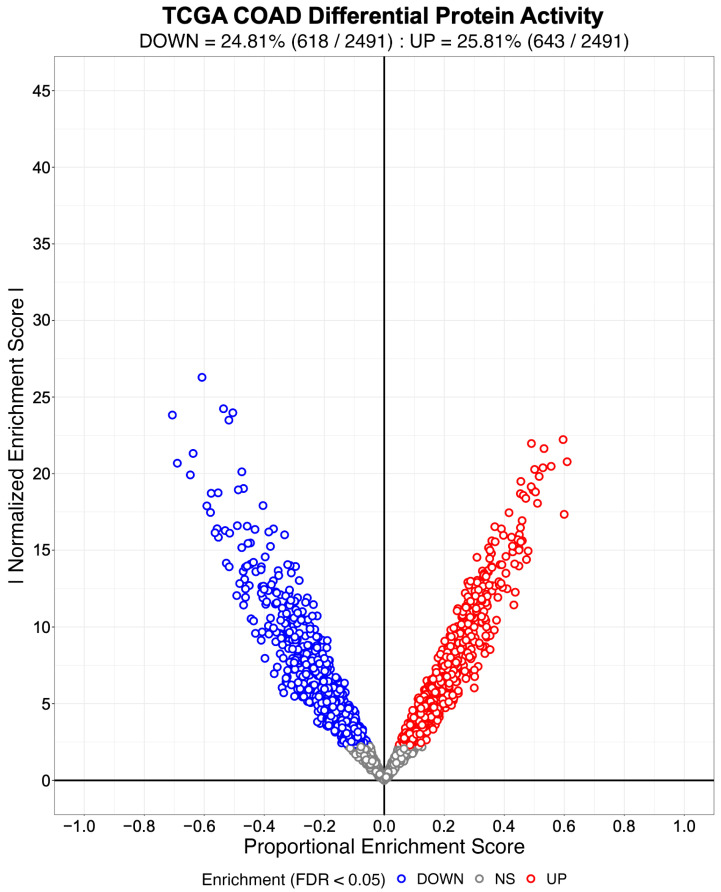
Differential protein activity volcano plot for the NaRnEA-inferred enrichment of TS-regulons in the differential gene expression signatures computed between primary tumors and phenotype-matched normal tissue samples from The Cancer Genome Atlas (TCGA) cohort for colon adenocarcinoma (COAD).

**Figure 3 entropy-25-00542-f003:**
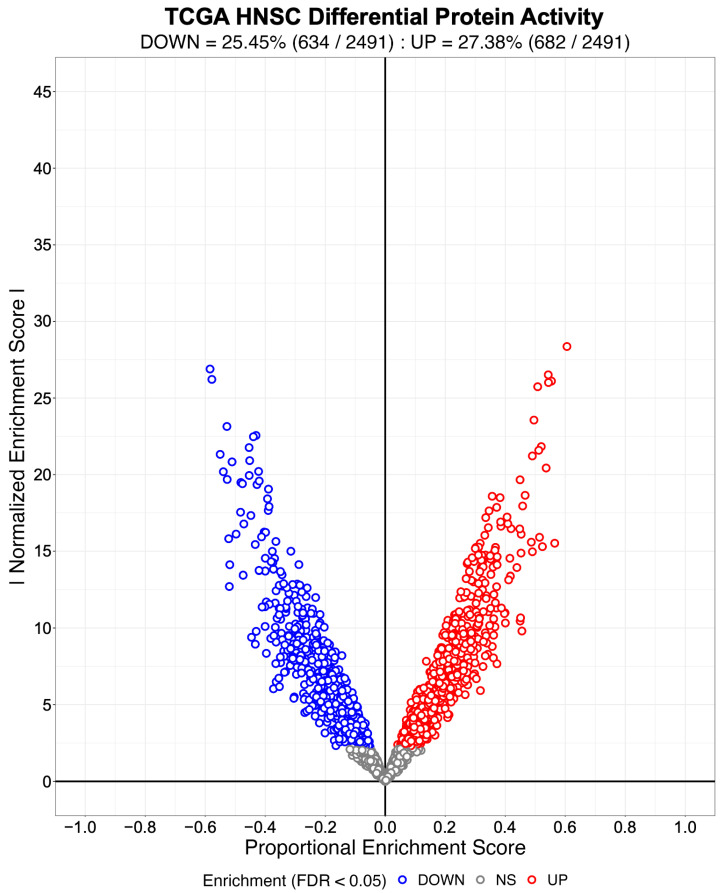
Differential protein activity volcano plot for the NaRnEA-inferred enrichment of TS-regulons in the differential gene expression signatures computed between primary tumors and phenotype-matched normal tissue samples from The Cancer Genome Atlas (TCGA) cohort for head and neck squamous cell carcinoma (HNSC).

**Figure 4 entropy-25-00542-f004:**
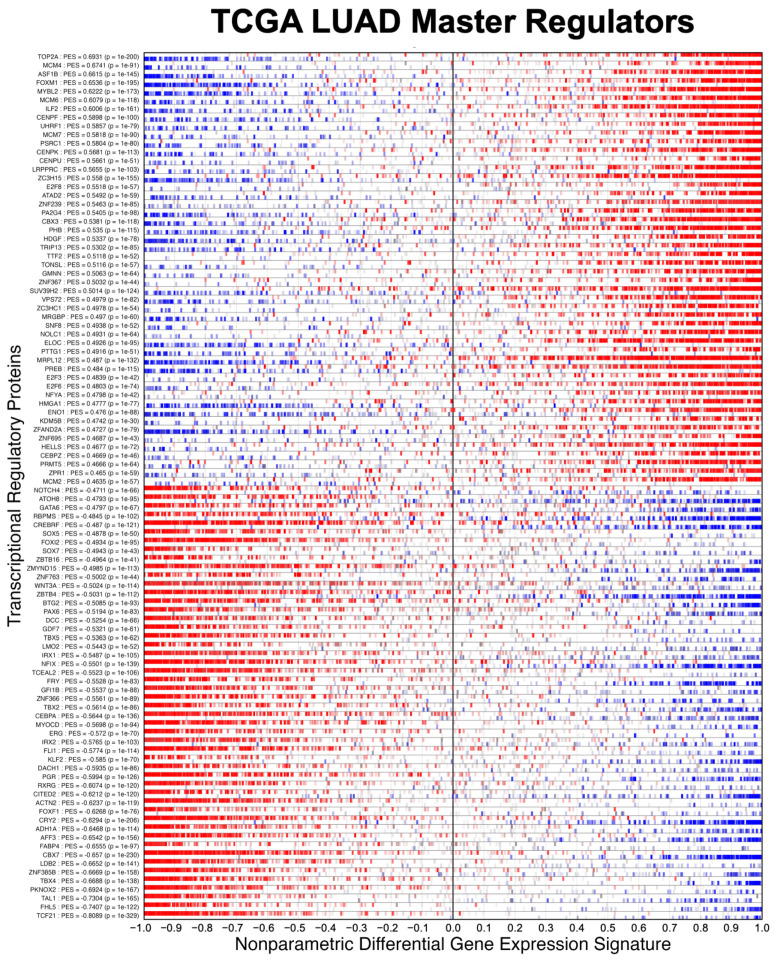
Master Regulator Analysis plot for the NaRnEA-inferred enrichment of the TS-regulons in the differential gene expression signatures computed between primary tumors and phenotype-matched normal tissue samples from The Cancer Genome Atlas (TCGA) cohort for lung adenocarcinoma (LUAD). The 50 most positively enriched and 50 most negatively enriched TS-regulons are selected for visualization based on the NaRnEA-inferred Proportional Enrichment Score (PES).

**Figure 5 entropy-25-00542-f005:**
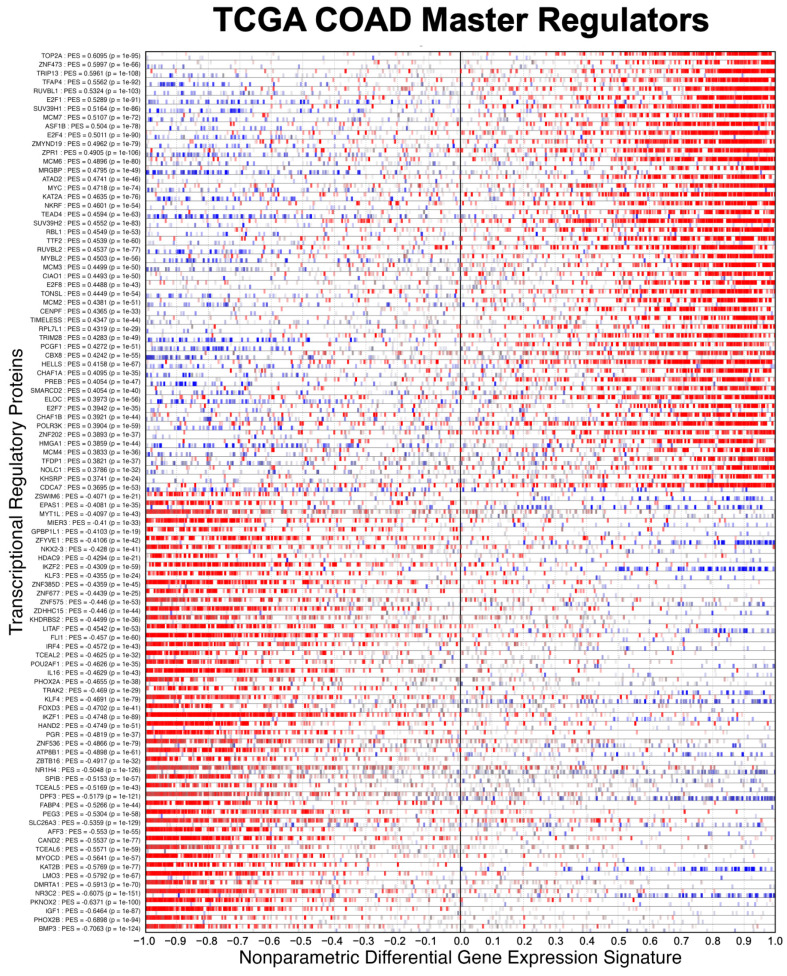
Master Regulator Analysis plot for the NaRnEA-inferred enrichment of the TS-regulons in the differential gene expression signatures computed between primary tumors and phenotype-matched normal tissue samples from The Cancer Genome Atlas (TCGA) cohort for colon adenocarcinoma (COAD). The 50 most positively enriched and 50 most negatively enriched TS-regulons are selected for visualization based on the NaRnEA-inferred Proportional Enrichment Score (PES).

**Figure 6 entropy-25-00542-f006:**
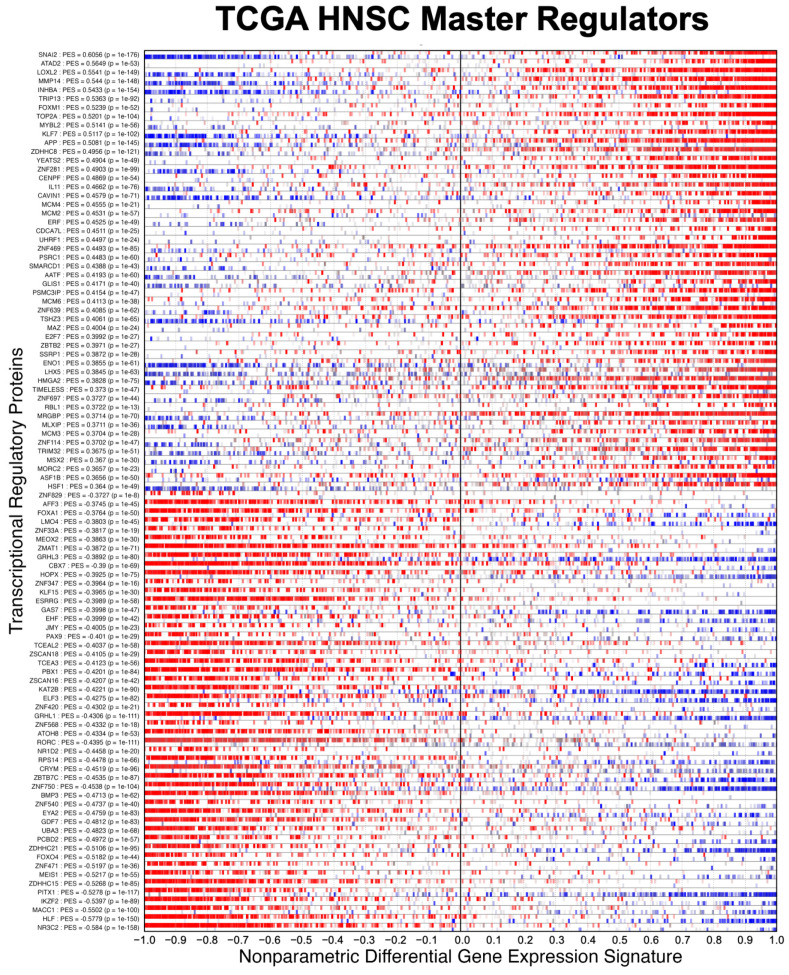
Master Regulator Analysis plot for the NaRnEA-inferred enrichment of the TS-regulons in the differential gene expression signatures computed between primary tumors and phenotype-matched normal tissue samples from The Cancer Genome Atlas (TCGA) cohort for head and neck squamous cell carcinoma (HNSC). The 50 most positively enriched and 50 most negatively enriched TS-regulons are selected for visualization based on the NaRnEA-inferred Proportional Enrichment Score (PES).

**Figure 7 entropy-25-00542-f007:**
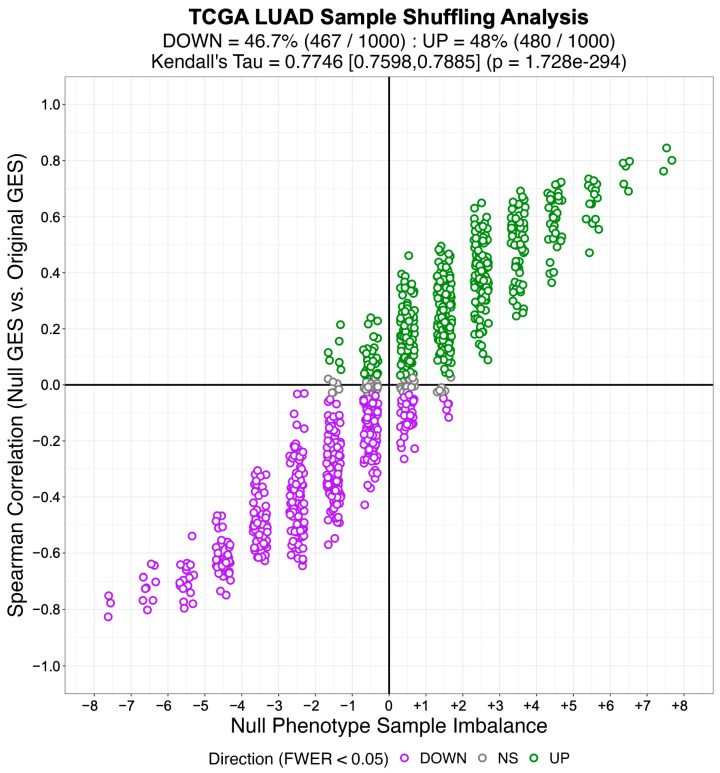
Statistically significant Spearman correlation between null differential gene expression signatures (GES) and the original differential gene expression signature (GES) computed between primary tumors and phenotype-matched normal tissue samples from The Cancer Genome Atlas (TCGA) cohort for lung adenocarcinoma (LUAD) are associated with, but not fully explained by, null phenotype sample imbalance.

**Figure 8 entropy-25-00542-f008:**
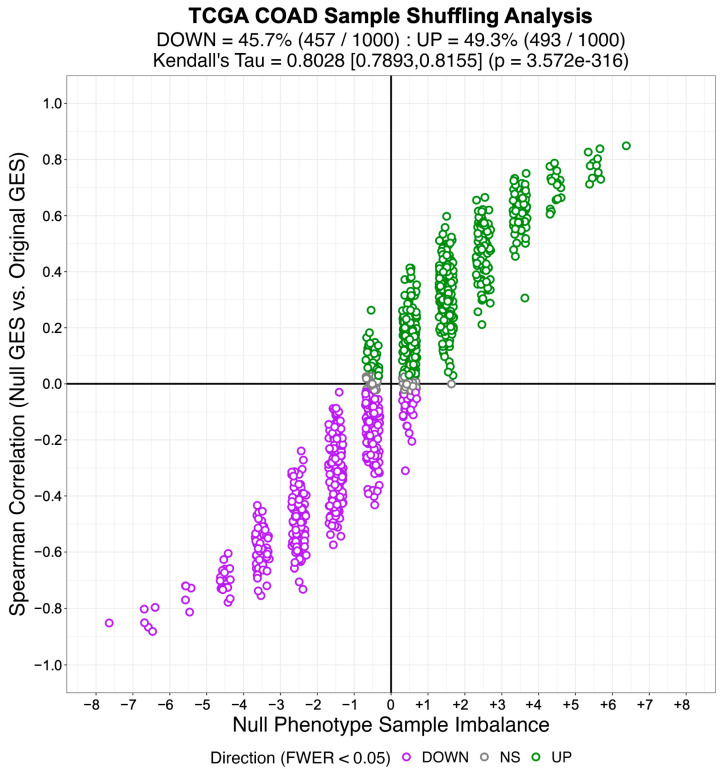
Statistically significant Spearman correlation between null differential gene expression signatures (GES) and the original differential gene expression signature (GES) computed between primary tumors and phenotype-matched normal tissue samples from The Cancer Genome Atlas (TCGA) cohort for colon adenocarcinoma (COAD) are associated with, but not fully explained by, null phenotype sample imbalance.

**Figure 9 entropy-25-00542-f009:**
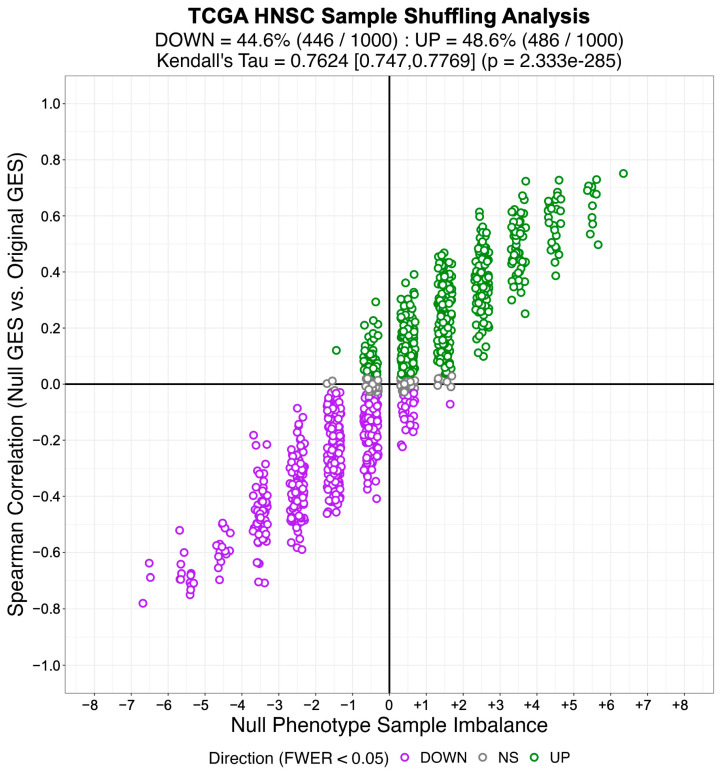
Statistically significant Spearman correlation between null differential gene expression signatures (GES) and the original differential gene expression signature (GES) computed between primary tumors and phenotype-matched normal tissue samples from The Cancer Genome Atlas (TCGA) cohort for head and neck squamous cell carcinoma (HNSC) are associated with, but not fully explained by, null phenotype sample imbalance.

**Table 1 entropy-25-00542-t001:** Proportion of NM regulons inferred to be statistically significantly enriched in differential gene expression signatures computed between primary tumor and adjacent normal tissue samples in TCGA LUAD, TCGA COAD, and TCGA HNSC by NaRnEA.

Type I Error Rate	TCGA LUAD NaRnEA-Inferred NM Regulon Enrichment	TCGA COAD NaRnEA-Inferred NM Regulon Enrichment	TCGA HNSC NaRnEA-Inferred NM Regulon Enrichment
FPR < 0.05	4.86% [4.05%, 5.78%]	5.50% [4.64%, 6.47%]	5.34% [4.49%, 6.30%]
FDR < 0.05	0% [0%, 0.15%]	0% [0%, 0.15%]	0% [0%, 0.15%]
FWER < 0.05	0% [0%, 0.15%]	0% [0%, 0.15%]	0% [0%, 0.15%]

**Table 2 entropy-25-00542-t002:** Proportion of TS regulons inferred to be statistically significantly enriched in differential gene expression signatures computed between primary tumor and adjacent normal tissue samples in TCGA LUAD, TCGA COAD, and TCGA HNSC by NaRnEA.

Type I Error Rate	TCGA LUAD NaRnEA-Inferred TS Regulon Enrichment	TCGA COAD NaRnEA-Inferred TS Regulon Enrichment	TCGA HNSC NaRnEA-Inferred TS Regulon Enrichment
FPR < 0.05	59.86% [57.90%, 61.79%]	54.11% [52.13%, 56.09%]	55.56% [53.58%, 57.52%]
FDR < 0.05	58.53% [56.57%, 60.47%]	50.62% [48.64%, 52.60%]	52.83% [50.85%, 54.81%]
FWER < 0.05	42.83% [40.88%, 44.80%]	32.78% [30.92%, 34.64%]	34.40% [32.54%, 36.31%]

**Table 3 entropy-25-00542-t003:** Three-by-three contingency table comparing NaRnEA-inferred differential protein activity from TCGA LUAD (rows) with MWU-inferred differential protein abundance from CPTAC LUAD (columns). Each cell in the table displays the number of proteins observed (Obs), the number of proteins expected under the null hypothesis that the rows and columns are independent (Exp), a Z-score computed from the standardized residual between the observed and expected values in the cell (Z), and a Bonferroni-corrected estimate of the cell-specific Family-Wise Error Rate based on the statistical significance of the cell-specific Z-score (FWER).

TCGA LUAD vs. CPTAC LUAD Kendall’s Tau-B = 0.3832 [0.3393, 0.4271] Chi-Squared Test *p* = 7.481 × 10^−51^			
	CPTAC DOWN (MWU Test)	CPTAC NS (MWU Test)	CPTAC UP (MWU Test)
TCGA UP (NaRnEA)	Obs = 77 Exp = 154.35 Z = −9.891 FWER = 4.079 × 10^−22^	Obs = 91 Exp = 106.92 Z = −2.158 FWER = 0.2782	Obs = 279 Exp = 185.73 Z = 11.31 FWER = 1.037 × 10^−28^
TCGA NS (NaRnEA)	Obs = 142 Exp = 159.88 Z = −2.160 FWER = 0.2768	Obs = 131 Exp = 110.74 Z = 2.850 FWER = 3.393 × 10^−2^	Obs = 190 Exp = 192.38 Z = −0.2241 FWER = 1.000
TCGA DOWN (NaRnEA)	Obs = 204 Exp = 108.77 Z = 12.95 FWER = 2.179 × 10^−37^	Obs = 71 Exp = 75.34 Z = −0.5846 FWER = 1.000	Obs = 40 Exp = 130.89 Z = −12.67 FWER = 7.848 × 10^−36^

**Table 4 entropy-25-00542-t004:** Three-by-three contingency table comparing NaRnEA-inferred differential protein activity from TCGA COAD (rows) with MWU-inferred differential protein abundance from CPTAC COAD (columns). Each cell in the table displays the number of proteins observed (Obs), the number of proteins expected under the null hypothesis that the rows and columns are in-dependent (Exp), a Z-score computed from the standardized residual between the observed and expected values in the cell (Z), and a Bonferroni-corrected estimate of the cell-specific Family-Wise Error Rate based on the statistical significance of the cell-specific Z-score (FWER).

TCGA COAD vs. CPTAC COAD Kendall’s Tau-B = 0.2913 [0.2333, 0.3492] Chi-Squared Test *p* = 1.456 × 10^−18^			
	CPTAC DOWN (MWU Test)	CPTAC NS (MWU Test)	CPTAC UP (MWU Test)
TCGA UP (NaRnEA)	Obs = 104 Exp = 147.20 Z = −6.260 FWER = 3.465 × 10^−9^	Obs = 45 Exp = 54.40 Z = −1.722 FWER = 0.7656	Obs = 167 Exp = 114.40 Z = 8.013 FWER = 1.006 × 10^−14^
TCGA NS (NaRnEA)	Obs = 153 Exp = 149.06 Z = 0.6447 FWER = 1.000	Obs = 66 Exp = 55.09 Z = 2.179 FWER = 0.2637	Obs = 101 Exp = 115.85 Z = −2.170 FWER = 0.2701
TCGA DOWN (NaRnEA)	Obs = 111 Exp = 71.74 Z = 7.230 FWER = 4.360 × 10^−12^	Obs = 25 Exp = 26.51 Z = −0.2254 FWER = 1.000	Obs = 18 Exp = 55.75 Z = −7.445 FWER = 8.757 × 10^−13^

**Table 5 entropy-25-00542-t005:** Three-by-three contingency table comparing NaRnEA-inferred differential protein activity from TCGA HNSC (rows) with MWU-inferred differential protein abundance from CPTAC HNSC (columns). Each cell in the table displays the number of proteins observed (Obs), the number of proteins expected under the null hypothesis that the rows and columns are in-dependent (Exp), a Z-score computed from the standardized residual between the observed and expected values in the cell (Z), and a Bonferroni-corrected estimate of the cell-specific Family-Wise Error Rate based on the statistical significance of the cell-specific Z-score (FWER).

TCGA HNSC vs. CPTAC HNSC Kendall’s Tau-B = 0.3455 [0.2964, 0.3946] Chi-Squared Test *p* = 7.481 × 10^−51^			
	CPTAC DOWN (MWU Test)	CPTAC NS (MWU Test)	CPTAC UP (MWU Test)
TCGA UP (NaRnEA)	Obs = 43 Exp = 95.2 Z = −7.980 FWER = 1.320 × 10^−14^	Obs = 97 Exp = 125.46 Z = −3.825 FWER = 1.179 × 10^−3^	Obs = 270 Exp = 189.32 Z = 10.222 FWER = 1.429 × 10^−23^
TCGA NS (NaRnEA)	Obs = 95 Exp = 104.51 Z = −1.311 FWER = 1.000	Obs = 168 Exp = 137.70 Z = 4.085 FWER = 3.967 × 10^−4^	Obs = 187 Exp = 207.79 Z = −2.500 FWER = 0.112
TCGA DOWN (NaRnEA)	Obs = 117 Exp = 55.27 Z = 10.221 FWER = 1.433 × 10^−23^	Obs = 71 Exp = 72.83 Z = −0.206 FWER = 1.000	Obs = 50 Exp = 109.90 Z = −8.983 FWER = 2.364 × 10^−18^

**Table 6 entropy-25-00542-t006:** Proportion of NM regulons inferred to be statistically significantly enriched in differential gene expression signatures computed between primary tumor and adjacent normal tissue samples in TCGA LUAD, TCGA COAD, and TCGA HNSC by GSEA.

Type I Error Rate	TCGA LUAD GSEA-Inferred NM Regulon Enrichment	TCGA COAD GSEA-Inferred NM Regulon Enrichment	TCGA HNSC GSEA-Inferred NM Regulon Enrichment
FPR < 0.05	10.88% [9.58%, 12.17%]	13.01% [11.71%, 14.39%]	1.49% [1.05%, 2.04%]
FDR < 0.05	0% [0%, 0.15%]	0% [0%, 0.15%]	0% [0%, 0.15%]
FWER < 0.05	0% [0%, 0.15%]	0% [0%, 0.15%]	0% [0%, 0.15%]

**Table 7 entropy-25-00542-t007:** Proportion of NM regulons inferred to be statistically significantly enriched in differential gene expression signatures computed between primary tumor and adjacent normal tissue samples in TCGA LUAD, TCGA COAD, and TCGA HNSC by aREA.

Type I Error Rate	TCGA LUAD aREA-Inferred NM Regulon Enrichment	TCGA COAD aREA-Inferred NM Regulon Enrichment	TCGA HNSC aREA-Inferred NM Regulon Enrichment
FPR < 0.05	3.41% [2.73%, 4.20%]	4.18% [3.42%, 5.04%]	5.14% [4.30%, 6.08%]
FDR < 0.05	0% [0%, 0.15%]	0% [0%, 0.15%]	0% [0%, 0.15%]
FWER < 0.05	0% [0%, 0.15%]	0% [0%, 0.15%]	0% [0%, 0.15%]

**Table 8 entropy-25-00542-t008:** Proportion of TS regulons inferred to be statistically significantly enriched in differential gene expression signatures computed between primary tumor and adjacent normal tissue samples in TCGA LUAD, TCGA COAD, and TCGA HNSC by GSEA.

Type I Error Rate	TCGA LUAD GSEA-Inferred TS Regulon Enrichment	TCGA COAD GSEA-Inferred TS Regulon Enrichment	TCGA HNSC GSEA-Inferred TS Regulon Enrichment
FPR < 0.05	7.51% [6.50%, 8.61%]	6.62% [5.68%, 7.67%]	5.26% [4.42%, 6.21%]
FDR < 0.05	0% [0%, 0.15%]	0% [0%, 0.15%]	0% [0%, 0.15%]
FWER < 0.05	0% [0%, 0.15%]	0% [0%, 0.15%]	0% [0%, 0.15%]

**Table 9 entropy-25-00542-t009:** Proportion of TS regulons inferred to be statistically significantly enriched in differential gene expression signatures computed between primary tumor and adjacent normal tissue samples in TCGA LUAD, TCGA COAD, and TCGA HNSC by aREA.

Type I Error Rate	TCGA LUAD aREA-Inferred TS Regulon Enrichment	TCGA COAD aREA-Inferred TS Regulon Enrichment	TCGA HNSC aREA-Inferred TS Regulon Enrichment
FPR < 0.05	10.88% [9.68%, 12.17%]	3.49% [2.81%, 4.29%]	6.62% [5.68%, 7.67%]
FDR < 0.05	0% [0%, 0.15%]	0% [0%, 0.15%]	0% [0%, 0.15%]
FWER < 0.05	0% [0%, 0.15%]	0% [0%, 0.15%]	0% [0%, 0.15%]

**Table 10 entropy-25-00542-t010:** Proportion of TN regulons inferred to be statistically significantly enriched in differential gene expression signatures computed between primary tumor and adjacent normal tissue samples in TCGA LUAD, TCGA COAD, and TCGA HNSC by NaRnEA.

Type I Error Rate	TCGA LUAD NaRnEA-Inferred TN Regulon Enrichment	TCGA COAD NaRnEA-Inferred TN Regulon Enrichment	TCGA HNSC NaRnEA-Inferred TN Regulon Enrichment
FPR < 0.05	0% [0%, 0.15%]	0% [0%, 0.15%]	0% [0%, 0.15%]
FDR < 0.05	0% [0%, 0.15%]	0% [0%, 0.15%]	0% [0%, 0.15%]
FWER < 0.05	0% [0%, 0.15%]	0% [0%, 0.15%]	0% [0%, 0.15%]

**Table 11 entropy-25-00542-t011:** Proportion of TN regulons inferred to be statistically significantly enriched in differential gene expression signatures computed between primary tumor and adjacent normal tissue samples in TCGA LUAD, TCGA COAD, and TCGA HNSC by GSEA using the alternative empirical gene-based permutation null model.

Type I Error Rate	TCGA LUAD GSEA-Inferred TN Regulon Enrichment	TCGA COAD GSEA-Inferred TN Regulon Enrichment	TCGA HNSC GSEA-Inferred TN Regulon Enrichment
FPR < 0.05	100% [99.85%, 100%]	100% [99.85%, 100%]	100% [99.85%, 100%]
FDR < 0.05	100% [99.85%, 100%]	100% [99.85%, 100%]	100% [99.85%, 100%]
FWER < 0.05	0% [0%, 0.15%]	0% [0%, 0.15%]	0% [0%, 0.15%]

## Data Availability

Publicly available datasets were analyzed in this study. All gene expression profiles from TCGA are available at (https://gdc.cancer.gov/ (accessed on 1 October 2020)). All protein abundance profiles from CPTAC are available at (http://linkedomics.org (accessed on 1 October 2020)). NaRnEA and ARACNe3 are freely available for research use on GitHub (https://github.com/califano-lab/NaRnEA (accessed on 3 March 2023)).

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
