# Peer review of "NaRnEA: An Information Theoretic Framework for Gene Set Analysis"

_entropy, 2023, doi:10.3390/e25030542_

Round 1
Reviewer 1 Report
The article proposes a novel non-parametric gene set enrichment method based on information theory. The authors provide a lengthy derivation of their method based on a null model created using maximum entropy principle. The authors claimed improved sensitivity of their proposed method compared with more traditional methods such as GSEA and aREA.
The paper has low readability and the derivation is hard to follow. A suggested approach would be to move most of the derivation to an appendix and highlight the key results and their implications in the main section of the article. Also the authors have used tables and text to suport their claims in the results section which is hard to comprehend. It would be recommended to include figures and plots in the results section.
Author Response
We acknowledge that the manuscript is long and highly mathematical in nature; however, we have endeavored wherever possible to render the derivation of the NaRnEA mathematical framework in a highly detailed way to lead the reader from our biochemical rationale through our information theoretic framework for gene set analysis to a novel null hypothesis significance test which we then evaluate in a systematic manner across three different datasets followed by straightforward validation. We note that the text and tables in our results section follow the conventions of other manuscripts which benchmark novel statistical methods for their control of the Type I and Type II error rate; additionally, we have included 9 figures to complement the text and tables and further guide the reader in addition to providing all code necessary for completely reproducing the results of this manuscript in the GitHub repository. Furthermore, all previous attempts to transfer critical mathematical details underlying the derivation of NaRnEA to an appendix and highlight only the salient points of our novel method have consistently led readers significantly astray from our intended communicatory goals; thus, we do not feel that is appropriate and we specifically selected the journal Entropy to allow for a longer and more mathematically-rich presentation of our method.
Reviewer 2 Report
This is a very interesting paper, well organized, well written and of high scientific quality. The only drawback is the extended length. I am sure the authors were aware of it, when they wrote the paper, and they already have considered options to make it smaller. However, I must say that a smaller paper is easier to read. I would suggest, but without this be a requirement, the authors consider again this issue, perhaps by creating an appendix.
Author Response
We thank the reviewer for their kind words regarding our new method. We note that all previous attempts to transfer critical mathematical details underlying the derivation of NaRnEA to an appendix and highlight only the salient points of our novel method have consistently led readers significantly astray from our intended communicatory goals; thus, we do not feel that is appropriate and we specifically selected the journal Entropy to allow for a longer and more mathematically-rich presentation of our method.
Reviewer 3 Report
It is a novel approach for a classic type of wide-spread analysis that overcomes some known limitations in GSEA and aREA, transforming complex biological mechanisms in a more realistic statistical model.
The paper itself might be complex to be understood for the average user of the proposed method, but its current form is appropriate for the given journal. I would suggest to provide a simplified use case in the main README of the Github repository to encourage its use.
Author Response
We thank the reviewer for their kind words regarding our new method. We have updated the README for the NaRnEA GitHub to provide a simplified use case to encourage the use of the algorithm (https://github.com/califano-lab/NaRnEA).
Author Response
We agree with the reviewer that simulation studies are extremely useful for benchmarking novel statistical methods, particularly with respect to establishing control of Type I error and Type II error rates. We note that our manuscript already leverages a form of simulation with the construction of null model (NM) regulons; since these regulons are constructed completely at random with no genuine transcriptional significance, they fulfill the Monte Carlo properties required for assessing control of the Type I error rate by NaRnEA, GSEA, and aREA. Thus, by using the NM-regulons we have achieved via simulation on the side of the gene sets that NaRnEA accurately controls the Type I error rate of gene set analysis. Crucially, we have done this while simultaneously leveraging real gene expression data rather than attempting to simulate gene expression data directly under either the null or alternative hypothesis since such an attempt would require the use of a chemical state equation which could best be described as a phenotype-specific stochastic partial differential equation encompassing >100,000 unique biomolecular species. We have also leveraged a form of simulation to demonstrate that the alternative null models for aREA and GSEA do not perform according to the specification of their creators the latter half of our Results section by shuffling gene expression signature identifiers several thousand times and evaluating the enrichment test statistics which result from the application of each gene set analysis method. We feel that these simulation studies have indeed enhanced the validity of the manuscript and that additional simulation studies are not feasible under current computational and scientific limitations.
The CPTAC database contains mass spectrometry proteomic data for cancer phenotypes and organ-matched normal tissue which correspond to the same phenotypes found in TCGA. The differential protein activity inferred by NaRnEA using ARACNe3-inferred regulon gene sets constructed from TCGA gene expression data cannot be validated with any ground truth since no biochemical method to directly measure transcriptional regulatory protein activity in vivo in a high-throughput, systematic manner has yet been constructed. Thus, the use of differential protein abundance data from CPTAC is an imperfect, but still highly informative, method of validating the NaRnEA-inferred differential protein activity using independent, experimental data. The VIPER method uses aREA to infer differential protein activity; in the Results section of our manuscript we explain that the VIPER package from Bioconductor is used to run aREA. Thus, VIPER cannot be used to validate NaRnEA because VIPER itself is simply a wrapper for aREA which is a method against which NaRnEA is being compared.
The Lindeberg-Central Limit Theorem provides guarantees for the asymptotic normality of the NaRnEA Normalized Enrichment Score under the null hypothesis of gene set analysis; we discuss in the derivation of NaRnEA that the rate of convergence for this asymptotic normality is governed by the extension of the Berry-Esseen theorem for non-identically distributed summands in the general case and that for well-balanced regulons we achieve excellent control of the Type I error rate with at least 30 targets. However, we note that the rate of convergence given by the Berry-Esseen theorem for non-identically distributed summands depends on the third central finite moment of the random variables which compose the summand and because of this the nonparametric transformation employed by NaRnEA is especially beneficial since it renders the resulting nonparametrically transformed differential gene expression signature values sub-Gaussian in their skewness under the null hypothesis of gene set analysis. Thus, for fewer than 30 targets in a gene set, the NaRnEA Normalized Enrichment Score is still rendered approximately normal and the tails of this distribution converge to the tails of the normal distribution from below; that is to say that for fewer than 30 targets, when tail probabilities are desired, especially tail probabilities obtained after correcting for multiple hypothesis testing, the Type I error rate is controlled conservatively in general. Thus, NaRnEA can in theory still be applied to gene sets with fewer than 30 targets, but strictly speaking the asymptotic normality is not completely guaranteed in this case; the extent to which the asymptotic normality is not guaranteed may be estimated by shuffling of gene expression signature identifiers and repeated application of NaRnEA to test for enrichment of a query gene set in the shuffled differential gene expression signatures in a manner similar to what was performed in the latter half of the Results section for the manuscript.
We have demonstrated the application of NaRnEA to three different cohorts of primary tumor samples as well as phenotype-matched adjacent normal tissue samples obtained from The Cancer Genome Atlas. We believe that given the high quality of these gene expression data and the fact that the differential protein activity inferred by NaRnEA from these data is validated with proteomic differential protein abundance data from phenotype-matched comparisons in CPTAC this provides strong validation for NaRnEA. However, we discuss in our updated Discussion section that Future Directions for this algorithm include its application to a broad array of phenotypes to further demonstrate its utility beyond cancer gene expression datasets.
Round 2
Reviewer 2 Report
I think the paper is at a scientific level, ready for publication
Reviewer 4 Report
Authors addressed the comments. Thank you!